# The Epstein-Barr virus deubiquitinating enzyme BPLF1 regulates the activity of topoisomerase II during productive infection

Jinlin Li[1☯¤a], Noemi Nagy[1☯], Jiangnan Liu[1], Soham Gupta[1¤b], Teresa Frisan[2], Thomas Hennig[3], Donald P. Cameron[1], Laura Baranello[1], Maria G. Masucci[1]*

**1** Department of Cell and Molecular Biology, Karolinska Institutet, Stockholm, Sweden, **2** Department of Molecular Biology, Umeå Center for Microbial Research, Umeå University, Umeå, Sweden, **3** Institute for Virology and Immunobiology, University of Würzburg, Würzburg, Germany

☯ These authors contributed equally to this work.
¤a Current address: Institute of Medical Biochemistry and Microbiology, Uppsala University, Uppsala, Sweden
¤b Current address: Division of Clinical Microbiology, Department of Laboratory Medicine Karolinska Institutet, Huddinge, Sweden.
* maria.masucci@ki.se

**Data Availability Statement:** All relevant data are within the manuscript and its Supporting Information files.

## Abstract

Topoisomerases are essential for the replication of herpesviruses but the mechanisms by which the viruses hijack the cellular enzymes are largely unknown. We found that topoisomerase-II (TOP2) is a substrate of the Epstein-Barr virus (EBV) ubiquitin deconjugase BPLF1. BPLF1 co-immunoprecipitated and deubiquitinated TOP2, and stabilized SUMOylated TOP2 trapped in cleavage complexes (TOP2ccs), which halted the DNA damage response to TOP2-induced double strand DNA breaks and promoted cell survival. Induction of the productive virus cycle in epithelial and lymphoid cell lines carrying recombinant EBV encoding the active enzyme was accompanied by TOP2 deubiquitination, accumulation of TOP2ccs and resistance to Etoposide toxicity. The protective effect of BPLF1 was dependent on the expression of tyrosyl-DNA phosphodiesterase 2 (TDP2) that releases DNA-trapped TOP2 and promotes error-free DNA repair. These findings highlight a previously unrecognized function of BPLF1 in supporting a non-proteolytic pathway for TOP2ccs debulking that favors cell survival and virus production.

## Author summary

The N-terminal domains of the herpesvirus large tegument proteins encode a conserved cysteine protease with ubiquitin- and NEDD8-specific deconjugase activity. Members of the viral enzyme family regulate different aspects of the virus life cycle including virus replication, the assembly of infectious virus particles and the host innate anti-viral response. However, only few substrates have been validated under physiological conditions of expression and very little is known on the mechanisms by which the enzymes contribute to the reprograming of cellular functions that are required for efficient infection and virus production. Cellular type I and type II topoisomerases (TOP1 and TOP2) resolve

**Funding:** This investigation was supported by grants awarded by the Swedish Cancer Society (CAN 2018/492) and the Medical Research Council (2019-101333) to M.G.M.. The work of SG and TH was partially supported by a grant awarded to the European ERA-NET eDEVILLI consortium. The funders had no role in study design, data collection and analysis, decision to publish, or preparation of the manuscript.

**Competing interests:** The authors have declared that no competing interests exist.

topological problems that arise during DNA replication and transcription and are therefore essential for herpesvirus replication. We report that the Epstein-Barr virus (EBV) ubiquitin deconjugase BPLF1 selectively regulates the activity of TOP2 in cells treated with the TOP2 poison Etoposide and during productive infection. Using transiently transfected and stable cell lines that express catalytically active or inactive BPLF1, we found that BPLF1 interacts with both TOP2α and TOP2β in co-immunoprecipitation and *in vitro* pull-down assays and the active enzyme stabilizes TOP2 trapped in TOP2ccs, promoting a shift towards TOP2 SUMOylation. This hinders the activation of DNA-damage responses and reduces the toxicity of Etoposide. The physiological relevance of this finding was validated using pairs of EBV carrying HEK-293T cells and EBV immortalized lymphoblastoid cell lines (LCLs) expressing the wild type or catalytic mutant enzyme. Using knockout LCLs we found that the capacity of BPLF1 to rescue of Etoposide toxicity is dependent on the expression of tyrosyl-DNA phosphodiesterase 2 (TDP2) that releases DNA-trapped TOP2 and promotes error-free DNA repair.

## Introduction

Epstein–Barr virus (EBV) is a human gamma-herpesvirus that establishes life-long persistent infections in most adults worldwide. The virus has been implicated in the pathogenesis of a broad spectrum of diseases ranging from infectious mononucleosis (IM) to a variety of lymphoid and epithelial cell malignancies including both Hodgkin and non-Hodgkin lymphomas, undifferentiated nasopharyngeal carcinoma, and gastric cancer [1].

Like other herpesviruses, EBV establishes latent or productive infections in different cell types. In latency, few viral genes are expressed resulting in the production of proteins and non-coding RNAs that drive virus persistence and cell proliferation [2]. In contrast, productive infection requires the coordinated expression of a large number of immediate early, early and late viral genes, which leads to the assembly of progeny virus and death of the infected cells [3]. Although much of the EBV-induced pathology has been attributed to viral latency, the importance of lytic products in the induction of chronic inflammation and malignant transformation is increasingly recognized [4, 5], pointing to inhibition of lytic gene products as a useful strategy for preventing EBV associated diseases.

EBV replication is triggered by the expression of immediate early genes, which transcriptionally activates a variety of viral and host cell factors required for subsequent phases of the productive cycle [6–8]. Among the cellular factors, DNA topoisomerase-I and -II (TOP1 and TOP2) were shown to be essential for herpesvirus DNA replication [9–11], raising the possibility that topoisomerase inhibitors may serve as antivirals. Indeed, non-toxic concentrations of TOP1 and TOP2 inhibitors were shown to suppress EBV-DNA replication [9], and different TOP1 inhibitors reduced the transcriptional activity of the EBV immediate-early protein BZLF1 and the assembly of viral replication complexes [12]. However, the mechanisms by which the virus harnesses the activity of these essential cellular enzymes remain largely unknown.

Topoisomerases sustain DNA replication, recombination and transcription by inducing transient single or double-strand DNA breaks that allow the resolution of topological problems arising from strand separation [13, 14]. TOP2 homodimers mediate DNA disentanglement by inducing transient double strand-breaks (DSBs) through the formation of enzyme-DNA adducts, known as TOP2 cleavage complexes (TOP2ccs), between catalytic tyrosine residues and the 5'ends of the DSBs [15]. Following the passage of the second DNA strand, TOP2

rejoins the DNA ends via reversion of the trans-esterification reaction. While TOP2-induced DSBs are relatively frequent in genomic DNA [16], failure to resolve TOP2ccs, as may occur upon endogenous or chemical stress that inhibits TOP2 activity, results in the formation of stable TOP2-DNA adducts that hinder DNA replication and transcription and trigger apoptotic cell death [17]. Thus, cellular defense mechanisms attempt to resolve the TOP2ccs via proteolytic or non-proteolytic pathways [18]. The proteolytic pathways involve the displacement of TOP2 via for example, ubiquitin [19] or SUMO and ubiquitin-dependent [20] proteasomal degradation, which, upon removal of residual peptide-DNA adducts by the Tyrosyl-DNA phosphodiesterase-2 (TDP2) resolving enzyme [21, 22], unmasks the DNA breaks and promotes the activation of DNA damage responses (DDR) [22]. Alternatively, the SUMOylation of TOP2 may induce conformational changes in the TOP2 dimer that exposes the covalent TOP2-DNA bonds to the direct action of TDP2 [23] without need for TOP2 proteolysis, which allows the repair of DSBs by TOP2 itself or other ligases. Two TOP2 isozymes expressed in mammalian cells share ~70% sequence identity and have similar catalytic activities and structural features but are differentially regulated and play distinct roles in biological processes [15]. While TOP2α is preferentially expressed in dividing cells and is essential for the decatenation of intertwined sister chromatids during mitosis [24], TOP2β is the only topoisomerase expressed in non-proliferating cells and is indispensable for transcription [25, 26].

Ubiquitin-specific proteases, or deubiquitinating enzymes (DUBs), regulate protein turnover by disassembling poly-ubiquitin chains that target the substrate for proteasomal degradation [27]. Several human and animal viruses encode DUB homologs that play important roles in the virus life cycle by promoting viral genome replication and inhibiting the host antiviral response [28–31]. In this study, we report that TOP2 is a substrate of the DUB encoded in the N-terminal domain of the EBV large tegument protein BPLF1 and provide evidence for the capacity of the viral enzyme to promote the non-proteolytic TDP2-dependent resolution of TOP2ccs, which enhances cell survival and favors virus production.

## Results

### BPLF1 selectively inhibits the degradation of TOP2 in cells treated with topoisomerase poisons

To investigate whether the EBV encoded DUB regulates the proteasomal degradation of poisoned topoisomerases, FLAG-tagged versions of the N-terminal catalytic domain of BPLF1 that is generated by caspase I cleavage of the large tegument protein during productive infection [32], and an inactive mutant where the catalytic Cys61 was substituted with Ala (BPLF1$^{C61A}$) were stably expressed by lentivirus transduction in HEK-293T cells under the control of a Tet-on regulated promoter (HEK-rtTA-BPLF1/BPLF1$^{C61A}$ cell lines). Inducible expression was monitored by probing immunoblots of cells treated for 24 h with increasing concentration of doxycycline (Dox) with antibodies to the FLAG or V5 tags (S1A Fig). Although the steady-state levels of BPLF1$^{C61A}$ were occasionally lower, both versions of the enzyme were readily detected by anti-FLAG immunofluorescence in more than 50% of the induced cells (S1B Fig).

To monitor ubiquitin-dependent proteasomal degradation, HEK-rtTA-BPLF1/BPLF1$^{C61A}$ cells cultured overnight in the presence or absence of Dox were treated with the TOP1 poison Camptothecin (Cpt) or the TOP2 poison Etoposide (Eto) in the presence or absence of the proteasome inhibitor MG132, and topoisomerase levels were assessed by western blot. Camptothecin and Etoposide trap TOP1-DNA and TOP2-DNA covalent adducts, respectively [33], while MG132 prevents the proteasomal degradation of stalled topoisomerase-DNA intermediates [19]. As expected, TOP1 was efficiently degraded in control Camptothecin treated cells

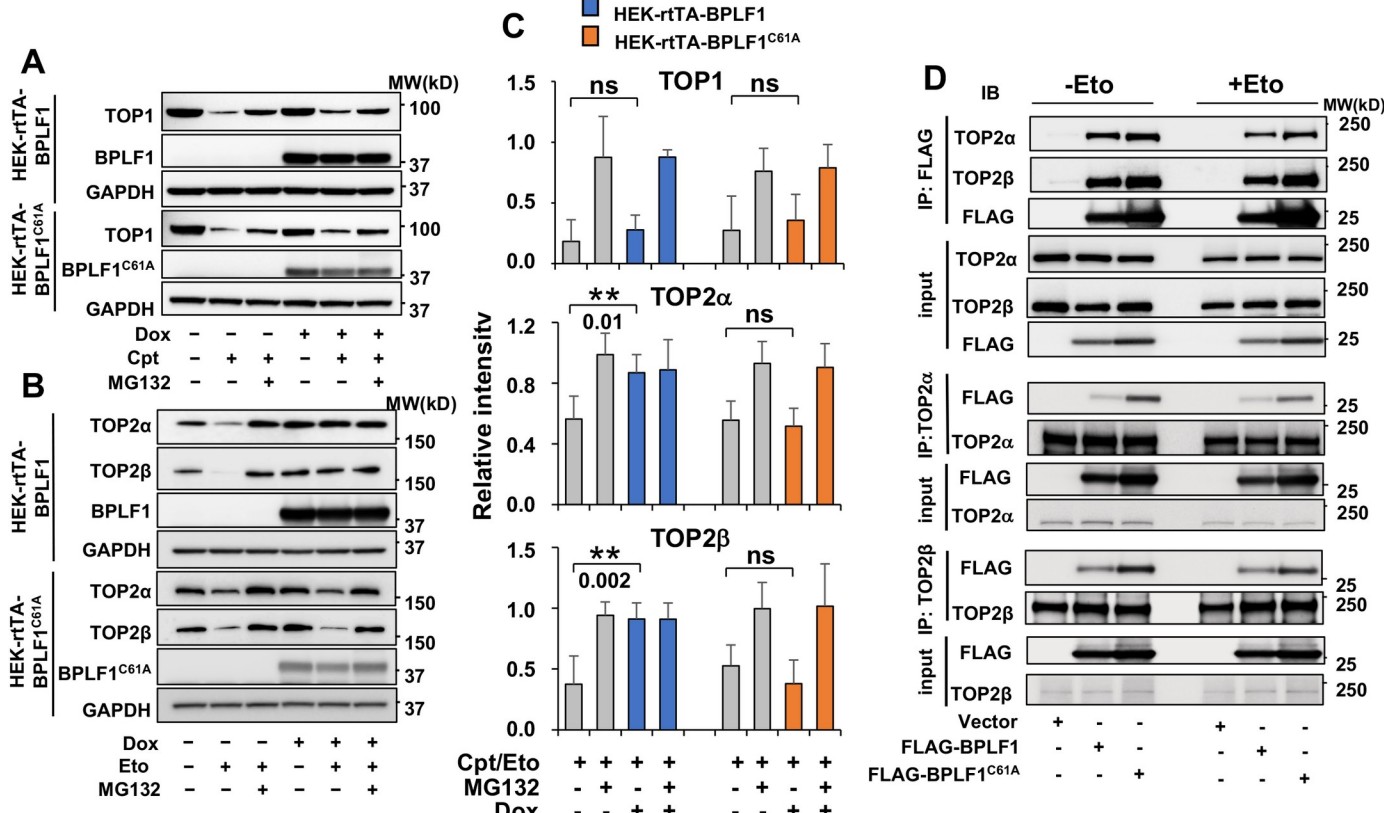

**Fig 1. BPLF1 selectively binds to TOP2 and inhibits the degradation of TOP2 in cells treated with topoisomerase poisons.** HEK-293T cell expressing inducible FLAG-BPLF1 or FLAG-BPLF1$^{C61A}$ were seeded into 6 well plates and treated with 1.5 µg/ml Dox for 24 h. After treatment for 3 h with 5 µM of the TOP1 poison Camptothecin (Cpt) or 6 h with 40 µM of the TOP2 poison Etoposide (Eto) with or without the addition of 10 µM MG132, protein expression was analyzed in western blots probed with the indicated antibodies. GAPDH was used as the loading control. (A) Representative western blots illustrating the expression of TOP1 in control and Cpt treated cells. The proteasome-dependent degradation of TOP1 induced by the treatment was not affected by the expression of BPLF1 or BPLF1$^{C61A}$ in Dox treated cells. (B) Representative western blots illustrating the expression of TOP2α and TOP2β in Etoposide treated cells. Expression of BPLF1 protected TOP2α and TOP2β from Etoposide-induced proteasomal degradation while BPLF1$^{C61A}$ had no appreciable effect. (C) The intensity of the TOP1, TOP2α and TOP2β specific bands in 5 (TOP1) or 6 (TOP2α and TOP2β) independent experiments was quantified using the ImageJ software. The data are presented as intensity of the bands in Cpt/Eto treated samples relative to untreated control after normalization to the GAPDH loading control. Statistical analysis was performed using Student's t-test. **P ≤ 0.01; ns, not significant. (D) HEK293T cells transfected with FLAG-BPLF1, FLAG-BPLF1$^{C61A}$, or FLAG-empty vector were treated with 40 µM Etoposide for 30 min and cell lysates were either immunoprecipitated with anti-FLAG conjugated agarose beads or incubated for 3 h with anti-TOP2α or TOP2β antibodies followed by the capture of immunocomplexes with protein-G coated beads. Catalytically active and inactive BPLF1 co-immunoprecipitate with both TOP2α and TOP2β in untreated and Etoposide treated cells (upper panels). Conversely, TOP2α (middle panels) and TOP2β (lower panels) interact with both catalytically active and inactive BPLF1. Representative western blots from one of two independent experiments where all conditions were tested in parallel are shown.

(Fig 1A and 1C upper panels), while treatment with Etoposide promoted the degradation of both TOP2α and TOP2β (Fig 1B and 1C middle and lower panels). The degradation was inhibited by treatment with MG132, confirming the involvement of the proteasome in the clearance of poisoned topoisomerases. Expression of catalytically active or mutant BPLF1 following Doxycycline treatment did not affect the Camptothecin-induced degradation of TOP1. In contrast, expression of the active BPLF1 was accompanied by stabilization of both TOP2α and TOP2β in Etoposide-treated cells with effect comparable to that induced by treatment with MG132. The mutant BPLF1$^{C61A}$ had no effect (Fig 1B and 1C). The selective rescue of the DNA-trapped TOP2 isozymes indicates that the effect cannot be ascribed to a global deubiquitination of cellular substrates by the overexpressed viral enzyme.

## TOP2 is a BPLF1 substrate

To directly test whether the TOP2 isozymes are substrates of BPLF1, we first investigated whether they interact in cells and in pull-down assays performed with recombinant proteins. Lysates of HEK-293T cells transiently transfected with FLAG-BPLF1 or FLAG-BPLF1$^{C61A}$ were immunoprecipitated with antibodies recognizing FLAG, TOP1, TOP2α or TOP2β. In line with the failure to rescue Camptothecin-induced degradation, BPLF1 did not interact with TOP1 (S2A Fig), whereas both TOP2α and TOP2β were readily detected in western blots of the FLAG immunoprecipitates and, conversely, BPLF1 was strongly enriched in the TOP2α and TOP2β immunoprecipitates indicating that the proteins interact in cells (Fig 1D). Notably, the failure to co-immunoprecipitate TOP1 and rescue TOP1 from Camptothecin-induced proteasomal degradation, supports the conclusion that the interaction of BPLF1 with TOP2 is not a mere artifact of overexpression. To gain insight on the nature of the interaction, equimolar concentration of yeast expressed FLAG-TOP2α, or a TOP2α mutant lacking the unique C-terminal domain that is not conserved in the TOP2β isozyme, FLAG-TOP2α-ΔCTD, were mixed with bacterially expressed His-BPLF1 and reciprocal pull-downs were performed with anti-FLAG (S2B Fig) or Ni-NTA coated beads (S2C Fig). A weak BPLF1 band was reproducibly detected in western blots of the FLAG-TOP2α pull-downs probed with a His-specific antibody and, conversely, a weak FLAG-TOP2α band was detected in the His pull-downs, confirming that the interaction is direct. The binding of BPLF1 to TOP2α was not affected by deletion of the TOP2α C-terminal domain (S2D Fig), pointing to the involvement of a domain shared by TOP2α and TOP2β in the direct binding of BPLF1 to the TOP2 isozymes. Notably, the weaker binding observed in the pull-down of bacterially expressed proteins compared to co-immunoprecipitation in cell lysates suggests that the interaction may be strengthened by factors, such as TOP2 post-translational modification or additional binding partners, that are only present in cells.

The capacity of BPLF1 to rescue TOP2 from proteasomal degradation together with the stronger interaction of TOP2 with the catalytically mutant BPLF1$^{C61A}$ (Fig 1D) point to TOP2 as a bona fide substrate of the viral enzyme. To investigate this possibility, TOP2α and TOP2β were immunoprecipitated from lysates of control and Etoposide-treated HEK-293T cells transiently transfected with BPLF1 or BPLF1$^{C61A}$ and western blots were probed with a ubiquitin-specific antibody. The cell lysates were prepared under denaturing conditions to exclude non-covalent protein interactions and working concentrations of NEM and iodoacetamide were added to all buffers to inhibit DUB activity. Transfection of the catalytically active BPLF1 appreciably reduced the total amount of ubiquitinated proteins in both untreated and Etoposide treated cells, (Fig 2A lower panels), confirming that the viral enzyme can deubiquitinate a broad range of cellular substrates. In line with the capacity of Etoposide to induce proteasomal degradation, smears of high molecular weight species corresponding to ubiquitinated TOP2α and TOP2β were detected in the immunoprecipitates of Etoposide-treated cells compared to untreated cells (Fig 2A). The intensity of the smears was strongly decreased in cells expressing active BPLF1, while the mutant BPLF1$^{C61A}$ had no appreciable effect. This, together with the selective rescue of TOP2 from proteasomal degradation, supports the conclusion that TOP2 is a true BPLF1 substrate.

The degradation of TOP2 by the proteasome plays an important role in the debulking of persistent TOP2ccs generated by topoisomerase poisons [19]. To investigate whether the viral DUB may interfere with this process, HEK-rtTA-BPLF1 cells cultured for 24 h in the presence or absence of Dox were treated for 30 min or 4 h with Etoposide with or without addition of MG132, and DNA-trapped TOP2α and TOP2β were detected by RADAR (rapid approach to DNA adduct recovery) assays [34, 35]. Neither TOP2α nor TOP2β were detected in control

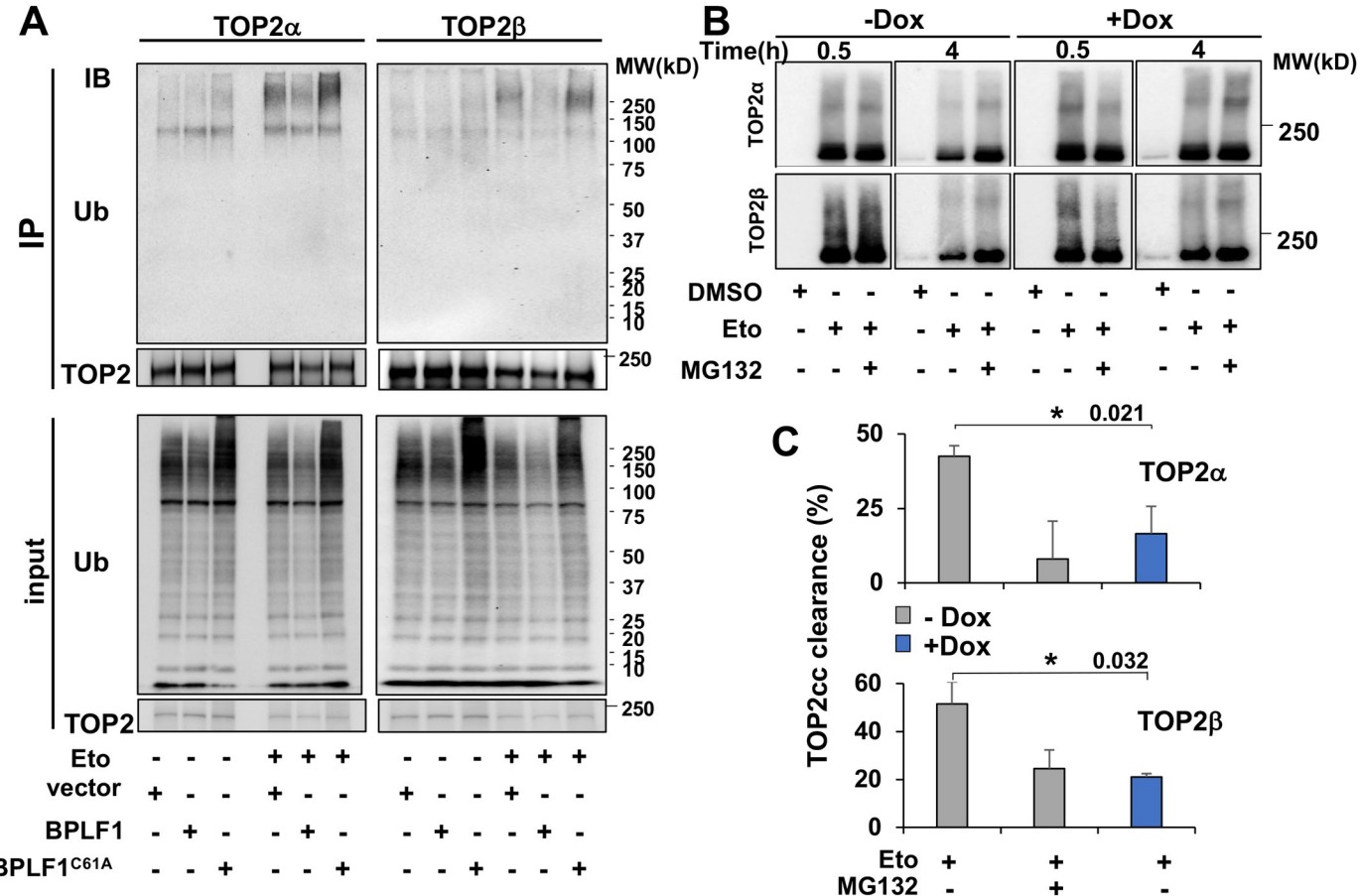

**Fig 2. BPLF1 deubiquitinates TOP2 and stabilizes TOP2ccs. (A)** HEK293T cells were transiently transfected with plasmids expressing FLAG-BPLF1, FLAG-BPLF1-C61A, or the FLAG empty vector, and aliquots were treated with 40 µM Etoposide for 30 min. TOP2α and TOP2β were immunoprecipitated from cell lysates prepared under denaturing conditions in the presence of DUB inhibitors and western blots were probed with antibodies to TOP2α, TOP2β and ubiquitin. The expression of catalytically active BPLF1 inhibits the ubiquitination of TOP2α and TOP2β induced by Etoposide treatment. Western blots from one representative experiment out of three are shown in the figure. **(B)** HEK-rtTA-BPLF1 cells were treated with 1.5 µg/ml Dox for 24 h followed by treatment with 80 µM Etoposide for the indicated time with or without the addition of 10 µM MG132. RADAR assays were performed as described in Materials and Methods and TOP2 trapped in 10 µg DNA was detected in western blots using antibodies to TOP2α or TOP2β. Trapped TOP2 appears as a major band of the expected size and a smear of higher molecular weight species. The intensity of the trapped TOP2α and TOP2β smears decreased over time in control untreated cells due to proteasomal degradation, while the decrease was significantly reduced upon expression of BPLF1 in Dox treated cells. Western blots from one representative experiment out of two are shown in the figure. **(C)** The intensity of the TOP2 smears was quantified using the ImageJ software. Clearance was calculated as 1-(intensity of the smears after treatment for 4 h/intensity of the smears after treatment for 30 min) x100. Treatment with MG132 reduced the clearance of TOP2ccs in BPLF1 negative cells and a similar reduction was achieved by expression of BPLF1 in Dox treated cells. The mean ± SD of two independent experiments is shown in the figure. Statistical analysis was performed using Student's t-test. *P≤ 0.05.

DMSO treated cells confirming that only covalently DNA-bound species are isolated by this method (Figs 2B and S3). In addition, preliminary experiments where control and Dox-treated cells were exposed to different concentration of Etoposide for 5–30 min showed that the expression of BPLF1 does not interfere with the formation of TOP2ccs (S3 Fig). In western blots of Etoposide treated samples, TOP2α and TOP2β appeared as major bands of the expected size and smears of high molecular weight species corresponding to various post-translational modifications. Despite minor experimental variations, comparable amounts of trapped TOP2α and TOP2β were detected in cells treated with Etoposide for 30 min, independently of BPLF1 expression or MG132 treatment (Fig 2B, compare 0.5 h Dox- versus Dox+), confirming that neither treatment, either alone or in combination, has significant effects on

the formation of TOP2ccs. As expected, in the absence of BPLF1 (Fig 2B, Dox- samples) the intensity of the TOP2 smears decreased after Etoposide treatment for 4 h. This was inhibited by MG132, confirming the involvement of proteasome-dependent degradation in the debulking of Etoposide-induced TOP2ccs. At the 4 h time point, the degradation of both TOP2α and TOP2β was significantly decreased in Dox treated cells (Fig 2B Dox- and Dox+ samples), corresponding to levels of stabilization comparable to those achieved by treatment with MG132. Quantification of the intensity of the TOP2 smears in repeated experiments confirmed that the expression of BPLF1 reduced the clearance of TOP2ccs as efficiently as treatment with MG132 (Fig 2C), supporting the conclusion that BPLF1 can deubiquitinate and stabilize TOP2 trapped in covalent DNA adducts. This finding was independently confirmed in experiments where TOP2ccs were stabilized by alkaline lysis [36] (S4A Fig). In this assay, smears of high molecular weight species were readily detected above the main TOP2β band in Dox-induced Etoposide-treated HEK-rtTA-BPLF1 cells, whereas smears were not detected when the blots were probed with a TOP1 specific antibody, confirming that the high molecular weight species correspond to DNA-trapped TOP2 (S4A Fig). As expected, the intensity of the smears decreased over time in BPLF1 negative cells, and the decrease was inhibited by MG132. In cells expressing catalytically active BPLF1, the intensity of the smears remained virtually constant over the observation time, resulting in significantly higher amounts of residual TOP2ccs (S4B Fig). Similar results were obtained when the blots were probed with antibodies to TOP2α.

## BPLF1 inhibits the detection of Etoposide-induced DNA damage and promotes cell survival

The proteolytic removal of DNA-trapped TOP2 exposes TOP2-induced DSBs that trigger the DDR. While limiting Etoposide toxicity, the DDR may also activate error-prone repair pathways that can cause genomic instability and apoptosis [19, 37, 38]. Inhibition of TOP2 degradation by treatment with MG132 was shown to prevent the activation of the DDR [19]. To test whether the capacity of BPLF1 to stabilize TOP2ccs affects the DDR, HEK-rtTA-BPLF1/BPLF$^{C61A}$ cells were cultured in the presence or absence of Dox for 24 h and then treated with Etoposide for 1 h, followed by staining for phosphorylated histone H2AX (γH2AX), a validated DDR marker [39]. As illustrated by representative fluorescence micrographs (Fig 3A, upper panels) and plots of γH2AX fluorescence intensity in BPLF1 positive and negative cells from the same slides (Fig 3B, upper panels), a diffuse γH2AX fluorescence was readily detected in Etoposide-treated BPLF1 negative cells and in cells expressing the mutant BPLF$^{C61A}$. Cells expressing active BPLF1 showed significantly weaker γH2AX fluorescence, suggesting that the viral enzyme inhibits the DDR. Accordingly, DNA repair was not triggered in BPLF1 positive cells as assessed by the impaired formation of 53BP1 foci that are visualized by immunofluorescence as small nuclear dots (Fig 3A and 3B, middle panels). A comparable BPLF1-dependent decrease of γH2AX fluorescence and impaired formation of 53BP1 and BRCA1 foci was observed upon Etoposide treatment in HeLa cells transiently transfected with BPLF1/BPLF1$^{C61A}$ (S5 Fig), confirming that the effect is not cell-type specific. To assess whether the failure to activate the DDR may be due to the capacity of BPLF1 to target events downstream of the formation and exposure DSBs, cells expressing BPLF1/BPLF1$^{C61A}$ were treated with the radiomimetic agent Neocarzinostain (NCS) [40]. Neither BPLF1 nor BPLF1$^{C61A}$ affected the induction of γH2AX in NCS treated cells (Fig 3A and 3B, lower panels). Collectively, these findings are consistent with a scenario where, by blocking the degradation of DNA-trapped TOP2, BPLF1 prevents the detection of TOP2-induced DSBs and avoids DDR activation.

Upon inhibition of TOP2 degradation, TOP2ccs may be resolved via a non-proteolytic pathway where SUMOylation-induced conformational changes in the TOP2 dimer expose the

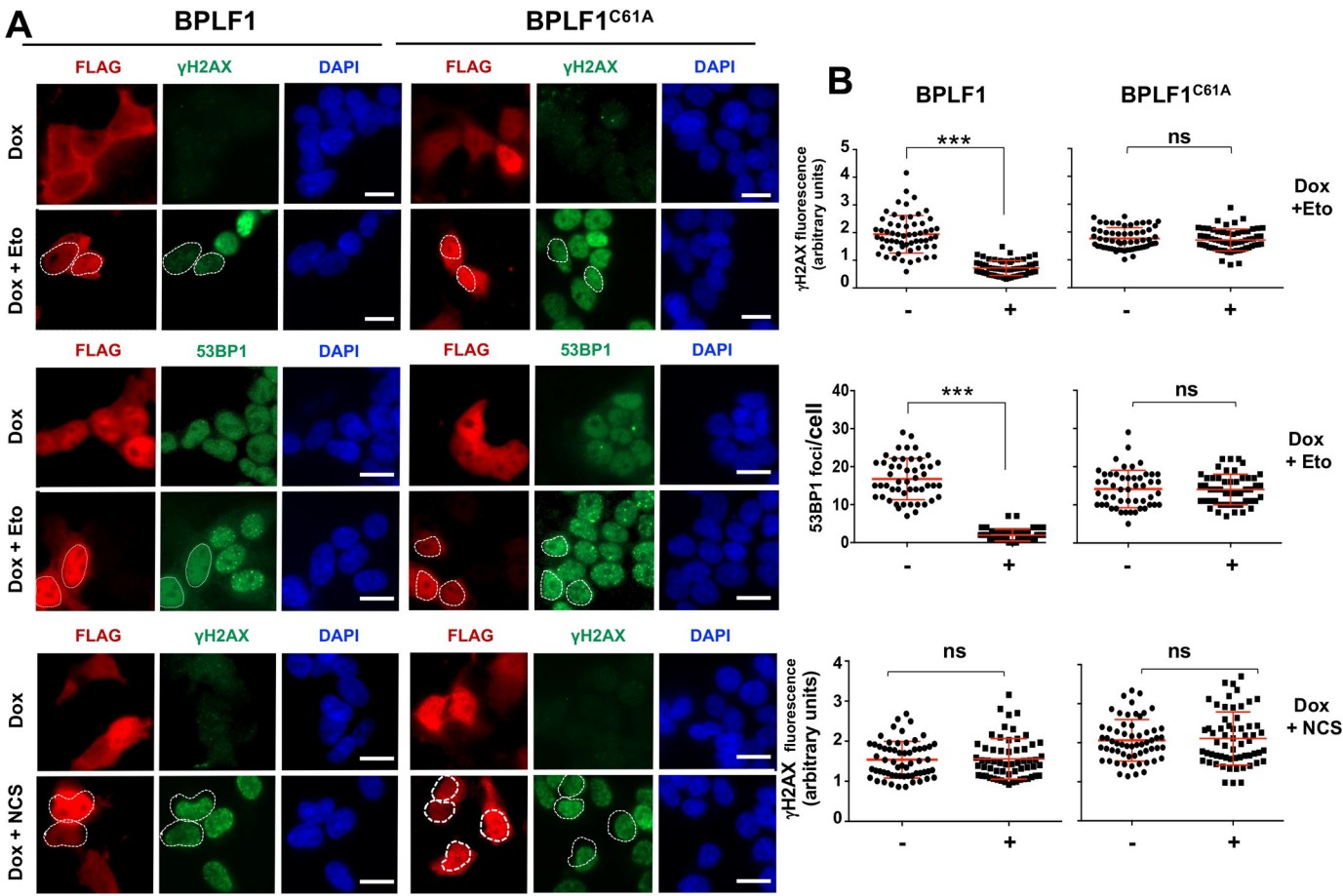

**Fig 3. BPLF1 selectively inhibits the detection of TOP2-induced DNA damage.** HEK-rtTA-BPLF1/BPLF1$^{C61A}$ cells grown on cover-slides were treated with 1.5 µg/ml Dox for 24 h to induce the expression of BPLF1 followed by treatment for 1 h with 40 µM Etoposide or 0.5 µg/ml of the radiomimetic Neocarzinostatin (NCS) before staining with the indicated antibodies. (**A**) The cells were co-stained with antibodies against FLAG (red) and antibodies to γH2AX or 53BP1 (green) and the nuclei were stained with DAPI (blue). The expression of catalytically active BPLF1 was associated with a significant decrease of Etoposide induced nuclear γH2AX fluorescence and decreased formation of 53BP1 foci. BPLF1$^{C61A}$ had no effect. Neither the catalytically active nor the inactive BPLF1 affected the induction of γH2AX in cells treated with NCS. Representative micrographs from one of two experiments where all conditions were tested in parallel are shown. Scale bar = 10 µm. (**B**) Quantification of γH2AX fluorescence intensity and number of 53BP1 foci in BPLF1/BPLF1$^{C61A}$ positive and negative cells from the same images. The Mean ± SD of fluorescence intensity or number of dots in at least 50 BPLF1-positive and 50 BPLF1-negative cells recorded in each condition is shown. Statistical analysis was performed using Student's t-test. ***P ≤0.001; ns, not significant.

tyrosyl-DNA bond to the catalytic site of Tyrosyl-DNA-phosphodiesterase-2 (TDP2) [23]. Thus, upon reversal of the covalent bond and release of the trapped TOP2, DSBs can be repaired without the need of potentially genotoxic nuclease resection [23, 38]. To assess whether this pathway may be engaged in BPLF1 expressing cells we first tested whether BPLF1 may affect the ubiquitination or SUMOylation of DNA-trapped TOP2. To this end, Dox-treated HEK-rtTA-BPLF1 cells were exposed to Etoposide for 30 min and western blots of TOP2ccs isolated by RADAR were probed with antibodies to ubiquitin and SUMO2/3. As expected, in control Etoposide treated cells the smears of high molecular weight species corresponding to DNA trapped TOP2 reacted with both the ubiquitin- and SUMO2/3-specific antibodies (Fig 4A). In contrast, while the intensity of the SUMO2/3 smear was largely unaffected, the extent of TOP2cc ubiquitination was strongly reduced in cells expressing the active viral DUB (Fig 4A upper and middle panels). This resulted in a relative enrichment of SUMOylated TOP2 (Fig 4B), which may favor the non-proteolytic debulking of TOP2ccs and counteract

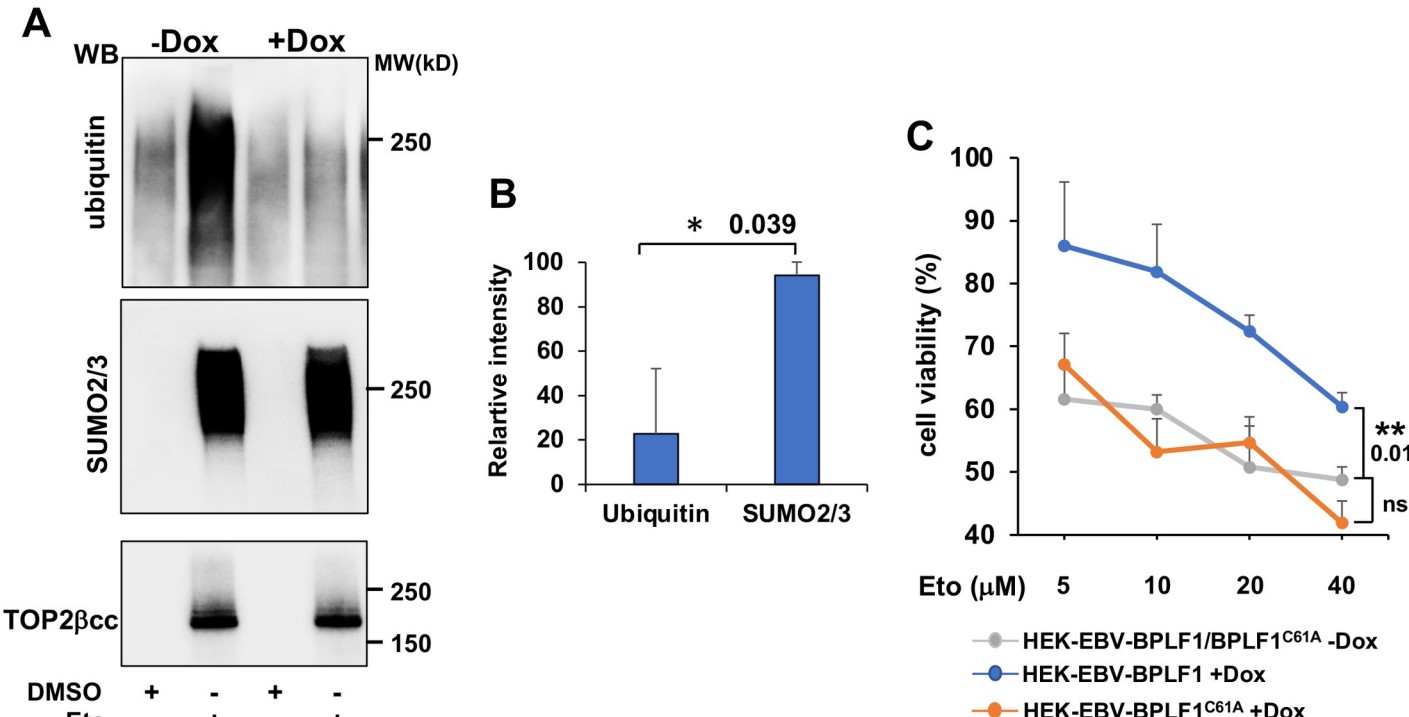

**Fig 4. BPLF1 promotes the accumulation of SUMOylated TOP2ccs and cell viability following Etoposide treatment.** (**A**) HEK-rtTA-BPLF1 cells were cultured for 24 h in the presence or absence of 1.5 μg/ml Dox and then treated with 80 μM Etoposide for 30 min followed by detection of DNA trapped TOP2 by RADAR assay. Western blots of proteins bound to 10 μg DNA were probed with antibodies to TOP2, ubiquitin and SUMO2/3. The expression of BPLF1 was associated with strongly decreased ubiquitination of the TOP2ccs while SUMOylation was only marginally affected. (**B**) The intensity of the ubiquitin/SUMO2/3 smears and TOP2 specific bands was quantified by densitometry using the ImageJ software. Relative intensity was calculated as intensity of the smears in Dox-treated versus untreated cells after normalization to the total amount of DNA-tapped TOP2. Mean ± SE of two independent experiments. Statistical analysis was performed using Student's t-test. $^*P \leq 0.05$. (**C**) HEK-rtTA-BPLF1/BPLF$^{C61A}$ cells were cultured for 24 h in the presence or absence of 1.5 μg/ml Dox and then treated overnight with the indicated concentration of Etoposide before assessing cell viability by MTT assays. The expression of catalytically active BPLF1 decreased the toxic effect of Etoposide over a wide range of concentrations while BPLF1$^{C61A}$ had no appreciable effect. The mean ± SD of two independent experiments is shown. Statistical analysis was performed using Student's t-test. $^{**}P \leq 0.01$; ns = non-significant.

the toxic effect of Etoposide. To test this possibility, cell viability was assessed by Thiazolyl blue tetrazolium bromide (MTT) assays in untreated and Dox-induced HEK-rtTA-BPLF1/BPLF1$^{C61A}$ cells following overnight exposure to increasing concentrations of Etoposide. Uninduced HEK-rtTA-BPLF1/BPLF1$^{C61A}$ were equally sensitive to Etoposide treatment whereas, in line with the hypothesized protective effect of BPLF1, the viability of Dox treated HEK-rtTA-BPLF1 cells was significantly improved over a wide range of Etoposide concentrations. The mutant BPLF1$^{C61A}$ had no appreciable effect (Fig 4C), supporting the conclusion that the DUB-dependent stabilization of TOP2 is instrumental for the capacity of BPLF1 to counteract Etoposide toxicity.

## BPLF1 regulates the activity of TOP2β during productive infection

In the next set of experiments, we asked whether physiological levels of BPLF1 regulate the activity of TOP2 in EBV infected cells. To this end, the productive virus cycle was induced in pairs of HEK293-EBV cells that carry recombinant EBV expressing wild type or mutant BPLF1 (HEK293-EBV-BPLF1/BPLF1$^{C61A}$) [41] and lymphoblastoid cell lines obtained by infecting normal B-lymphocytes with virus rescued from the HEK293-EBV cell lines (LCL-BPLF1/BPLF1$^{C61A}$). Since standard protocols for induction of the productive cycle work

poorly in these cells, all cell lines were stably transduced with a recombinant lentivirus expressing the viral transactivator BZLF1 under the control of a tetracycline-regulated promoter. Treatment with doxycycline induced the upregulation of early (BMRF1, DNA polymerase processivity factor) and late (BFRF3, capsid protein) lytic viral antigens detected in western blots probed with specific antibodies (S6A Fig), supporting their capacity to proficiently complete the productive cycle, and BMRF1 immunofluorescence confirmed that comparable numbers of cells entered the productive cycle. Since the available BPLF1 specific antibodies do not detect endogenous BPLF1, the expression of BPLF1 mRNA was assayed by qPCR (S6B Fig), which confirmed comparable levels of expression. Comparison of HEK293-EBV cell pair carrying wild type and mutant BPLF1 revealed that expression of the active enzyme promotes a small but reproducible increase of TOP2α and TOP2β protein levels in the absence of transcriptional upregulation (Figs 5A, 5B and S6C), stronger accumulation of TOP2α and TOP2β cleavage complexes (Fig 5C and 5D), and a significantly weaker activation of the DDR, as assessed by the intensity of γH2AX specific bands in western blots (Fig 5E and 5F). Thus, the key features of the phenotype observed in the BPLF1 inducible cell lines, i.e. stabilization of TOP2, accumulation of TOP2ccs and inhibition of the DDR, were reproduced upon expression of endogenous BPLF1 in productively infected cells.

In the LCLs, induction of the productive cycle was accompanied by a highly reproducible transcriptional downregulation of TOP2α with no significant changes in the expression of TOP2β (Figs 6A, 6B and S6C). The transcriptional downregulation of TOP2α is consistent with the previously reported BZLF1-induced establishment of a pseudo S-phase, characterized by G0/G1 arrest and inhibition of cellular DNA synthesis with concomitant activation of S-phase promoting cyclin E/A-CDK2 CDKs and accumulation of hyperphosphorylated Rb protein in cells with an intact p53 pathway [42]. This stands in sharp contrast with the BPLF1 deNEDDylase-dependent endoreduplication phenotype observed in EBV carrying cells with inactivated p53, such as the HEK293 cell line [30]. Despite the different behavior regarding TOP2α expression and consequent failure to detect TOP2αccs, as observed in HEK293-EBV cell line, the expression of catalytically active BPLF1 was associated with a significantly stronger accumulation of TOP2βccs in EBV immortalized LCLs (Fig 6C and 6D).

Since LCLs faithfully reproduce the physiological interaction of the virus with normal B lymphocytes, this cell model was used in subsequent experiments. To investigate whether the accumulation of TOP2ccs correlates with changes in the levels of TOP2 ubiquitination or SUMOylation, TOP2β was immunoprecipitated from lysates of induced LCL-BPLF1/BPLF1-C61A under denaturing conditions and western blots of the immunoprecipitates were probed with antibodies to ubiquitin and SUMO2/3. In accordance with our previous findings [32], physiological levels of the viral enzyme had no appreciable effects on the overall abundance of ubiquitinated proteins (Fig 6E upper panel, input) whereas a significant decrease of ubiquitinated TOP2β was reproducibly observed in cells expressing active BPLF1 (Fig 6E upper panel, TOP2β IP), confirming that TOP2β is a true BPLF1 substrate in virus infected cells. As previously reported [43], induction of the productive virus cycle was associated with the accumulation of poly-SUMOylated proteins detected by the SUMO2/3 specific antibody (Fig 6E, lower panel input), which is due to downregulation of the SUMO-dependent ubiquitin ligase RNF4 by viral miRNA expressed during productive infection. SUMOylated TOP2β was increased in cells expressing both catalytically active and inactive BPLF1, (Fig 6E, lower panel TOP2β IP), which shifted the SUMO/ubiquitin ratio towards SUMOylated TOP2 in cells expressing catalytically active BPLF1 (Fig 6F). As observed in the HEK293-EBV-BPLF1/BPLF1-C61A pair, induction of the productive cycle was associated with a significantly weaker activation of the DDR in cells expressing the active viral enzyme as assessed by intensity of the γH2AX specific band in western blot (Fig 6G and 6H). We then asked whether the

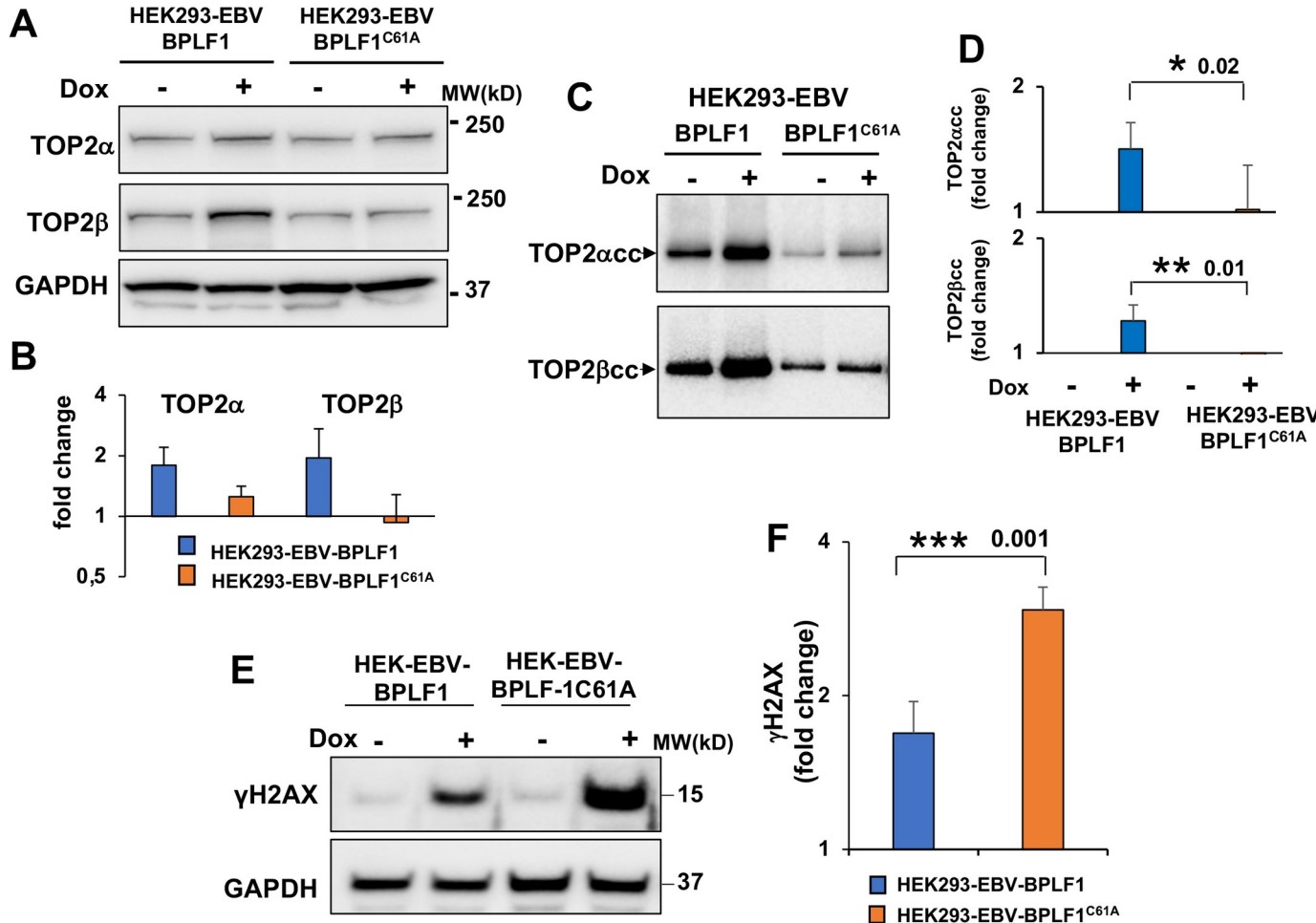

**Fig 5. BPLF1 regulates the activity of TOP2 in productively infected HEK293-EBV cells.** The productive virus cycle was induced by treatment with 1.5 µg/ml Dox in HEK293-EBV cells carrying recombinant EBV encoding wild type or catalytic mutant BPLF1 and a tetracycline regulated BZLF1 transactivator. The expression of TOP2α and TOP2β was assessed by western blot and the intensity of the specific bands was quantified using the ImageJ software. (**A**) Representative western blots illustrating the small upregulation of TOP2α and TOP2β in cells expressing catalytically active BPLF1. (**B**) Densitometry quantification of the specific bands was performed using the ImageJ software and the values were normalized to the GAPDH loading control. The mean ± SD of three independent experiments are shown. (**C**) Representative RADAR assay illustrating the accumulation of TOP2α and TOP2β cleavage complexes in cells expressing catalytically active BPLF1. Western blots from one representative experiment out of three are shown. (**D**) The intensity of the DNA trapped TOP2α and TOP2β species was quantified using the ImageJ software. The mean ± SD fold increase in induced versus non induced cells recorded in three independent experiments is shown. Statistical analysis was performed using Student's t-test. *P≤0.05; **P≤0.01. (**E**) Representative western blot illustrating the expression of the DDR marker γH2AX in induced HEK293-EBV cells expressing catalytic active or inactive BPLF1. (**F**) The intensity of γH2AX bands were quantified by densitometry in three independent experiments. The fold increase in induced versus control cells was calculated after normalization to the GAPDH loading control. Statistical analysis was performed using Student's t-test. ***P≤0.001.

BPLF1-induced changes in TOP2 ubiquitination/SUMOylation ratio, stabilization of TOP2ccs and inhibition of the DDR may impact on the sensitivity of induced LCLs to the toxic effect of Etoposide. As observed with the conditional HEK-rtTA-BPLF1/BPLF1$^{C61A}$ cell lines (Fig 4C), expression of the active viral DUB counteracted the toxic effect of Etoposide in induced LCLs over a wide range of Etoposide concentrations while the inactive enzyme lacked significant protective activity (Fig 6I).

In the final set of experiments, we further probed the relationship between the BPLF1-mediated stabilization of DNA-trapped TOP2 and Etoposide toxicity. To this end, we took advantage of the previously described EBV immortalized LCL TK6 and the subline TK6-TDP2$^{-/-}$ that lacks the TDP2 enzyme [44]. TDP2 facilitates the repair of TOP2-induced DSBs by

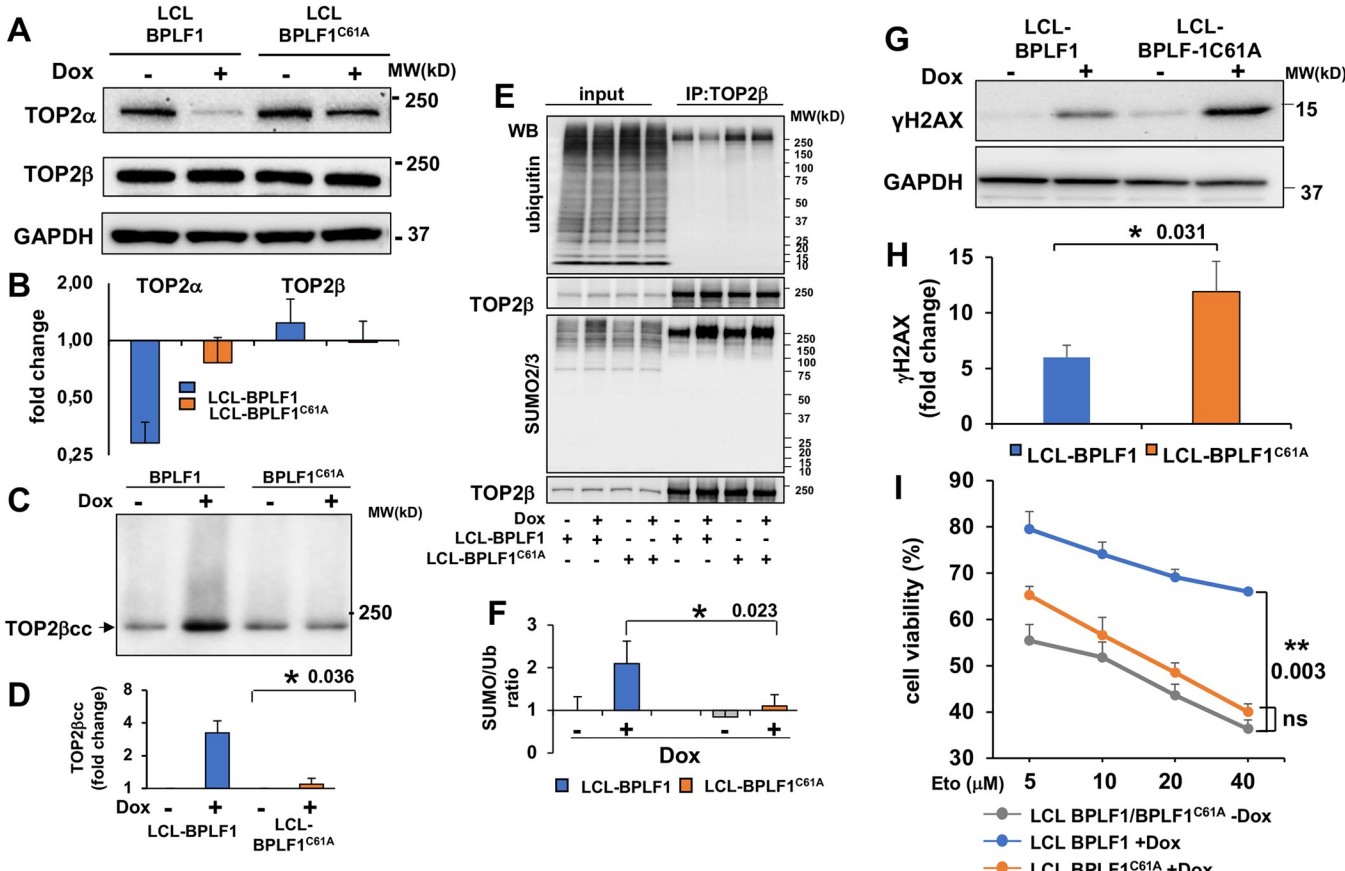

**Fig 6. BPLF1 regulates the expression and activity of TOP2 in productively infected LCLs.** The productive virus cycle was induced by treatment with 1.5 µg/ml Dox in LCL cells carrying recombinant EBV encoding wild type or catalytic mutant BPLF1 and a tetracycline regulated BZLF1 transactivator. Induction of the productive cycle was associated with a highly reproducible downregulation of TOP2α while TOP2β was either unchanged or slightly increased. The effect was stronger in cells expressing wild type BPLF1. (**A**) Representative western blots illustrating the expression of TOP2α and TOP2β in control and induced cells. (**B**) The intensity of the specific bands was quantified using the ImageJ in three to five independent experiments and fold change in induced versus control cells was calculated after normalization to the GAPDH loading control. (**C**) The formation of TOP2βccs was investigated by RADAR assays in untreated and induced LCLs. Representative western blot illustrating the significant increase of TOP2ccs upon induction of the productive virus cycle in LCL cells expressing catalytically active BPLF1. BPLF1$^{C61A}$ had no appreciable effect. One representative western blot is shown. (**D**) Quantification of the intensity of the TOP2β smears in three independent experiments. Fold increase was calculated as the ratio between the smear intensity in control versus induced cells. $^{*}P\leq0.05$. (**E**) TOP2β was immunoprecipitated from total cell lysates of control and induced LCLs and western blots were probed with antibodies to TOP2β, ubiquitin and SUMO2/3. Western blots illustrating the increase of SUMOylated TOP2β in productively infected cells and selective decrease of ubiquitinated TOP2β in cells expressing catalytically active BPLF1. One representative experiment out of three is shown. (**F**) The intensity of the bands corresponding to immunoprecipitated TOP2β and ubiquitinated or SUMOylated species was quantified using the ImageJ software and the SUMO/Ub ratio was calculated after normalization to immunoprecipitated TOP2β. The mean ± SE of three independent experiments is shown. $^{*}P<0.05$. (**G**) Representative western blot illustrating the expression of the DDR marker γH2AX in control and induced LCL-EBV-BPLF1/BPLF1$^{C61A}$ cells. (**H**) The intensity of γH2AX bands were quantified by densitometry in eight independent experiments. The fold increase in induced versus control cells was calculated after normalization to the GAPDH loading control and to the level of induction as assessed by the intensity of the BMRF1 specific band. Statistical analysis was performed using Student's t-test. $^{*}P\leq0.05$. (**I**) The productive cycle was induced in LCL-EBV-BPLF1/BPLF1$^{C61A}$ by culture for 72 h in the presence 1.5 µg/ml Dox. After washing and counting, $5\times10^{4}$ live cells were seeded in triplicate wells of 96 well plates and treated overnight with the indicated concentration of Etoposide before assessing cell viability by MTT assays. Catalytically active BPLF1 enhanced cell viability over a wide range of Etoposide concentration while BPLF1$^{C61A}$ had no appreciable effect. The mean ± SE of cell viability in three independent experiments is shown. $^{**}P\leq0.01$.

employing its 5′-tyrosyl DNA phosphodiesterase activity to hydrolyze the phosphotyrosyl bond between the poisoned topoisomerase and DNA, thereby liberating the 5′ termini of the DSB for DNA ligation [45, 46]. We reasoned that, if the capacity of BPLF1 to counteract the toxic effect of etoposide is mediated by inhibition of the proteolytic clearance of poisoned TOP2, reversal of the covalent TOP2-DNA bonds by TDP2 may be required for cell survival. To test this possibility, stable sublines of the TK6 and TK6-TDP2$^{-/-}$ LCLs expressing

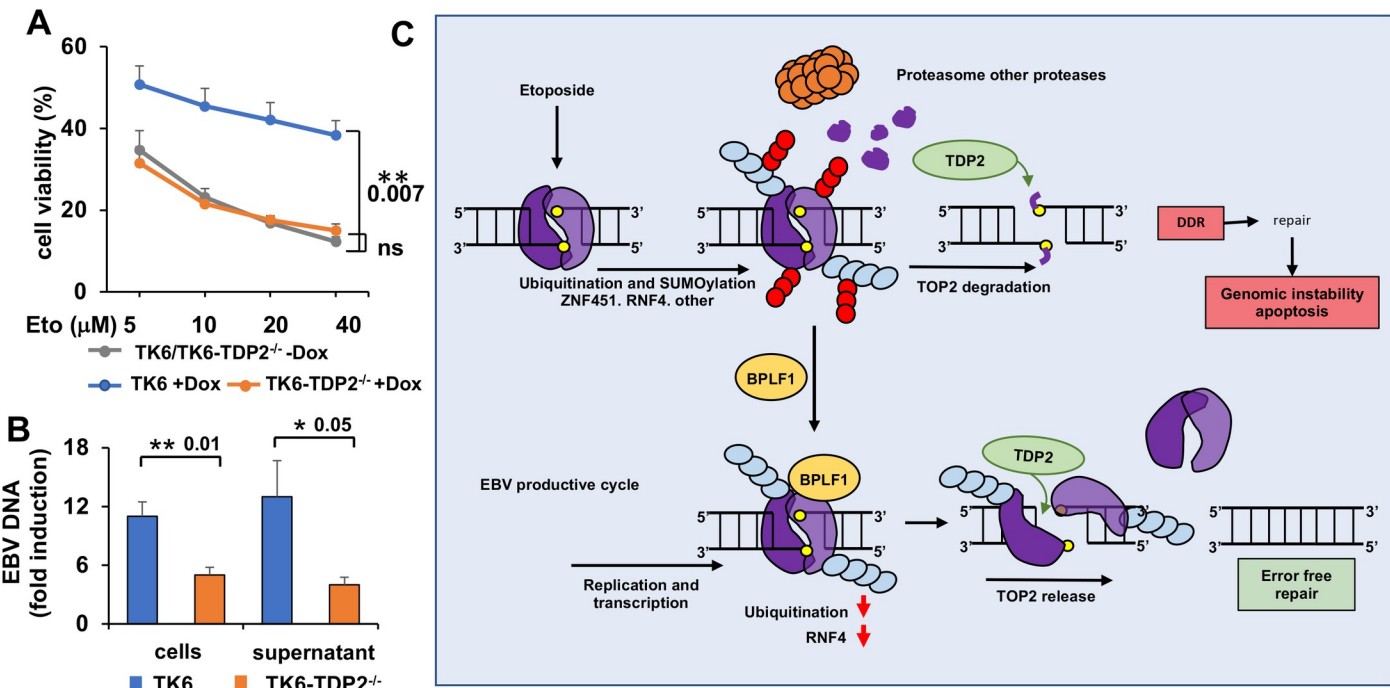

**Fig 7. TDP2 is required for resistance to Etoposide toxicity and virus production in induced LCLs.** The productive virus cycle was induced in TK6 and TK6-TDP2$^{-/-}$ LCLs expressing a tetracycline regulated BZLF1 transactivator by treatment with 1.5 µg/ml Dox. (**A**) TK6/TK6-TDP2$^{-/-}$ LCLs were harvested 72 h after induction. Five x10$^4$ live cells were seeded in triplicate wells of 96 well plates and treated overnight with the indicated concentration Etoposide before assessing cell viability by MTT assays. The rescue of Etoposide toxicity observed in induced TK6 was abolished in cells lacking TDP2. The mean ± SE of cell viability in three independent experiments is shown.$^{**}$P≤0.01. (**B**) The amount of EBV-DNA in cell pellets and DNAse treated culture supernatants was quantified by qPCR 72 h after induction. The mean ± SE fold induction relative to untreated controls in three (supernatants) or four (cells) independent experiments is shown. $^*$P≤0.05, $^{**}$P≤0.01. (**C**) Model of TOP2 regulation by BPLF1. TOP2 (violet) trapped in TOP2ccs (yellow) is targeted for proteasomal degradation via SUMOylation (light blue) and ubiquitination (red) mediated by the SUMO ligase ZNF451, the SUMO-targeting ubiquitin ligase RNF4 and other cellular ubiquitin ligases, leading to the display of partially digested 5'-phosphotyrosyl-DNA adducts. Processing by the TDP2 resolvase generates protein-free DSBs that trigger the DDR. Imprecise repair leads to apoptosis and genomic instability. Ectopically expressed BPLF1 is recruited to DNA-trapped TOP2 and inhibits proteasomal degradation, which prevents the activation of the DDR. In the absence of ubiquitination, SUMOylation may alter the conformation of the TOP2 dimer allowing direct access of TDP2 to the 5'-phosphotyrosyl-DNA bonds, which promotes error-free repair. During productive EBV infection, the concomitant expression of BPLF1 and viral miRNA mediated downregulation of RNF4 favors the accumulation of SUMOylated TOP2β and the activation of non-proteolytic pathways for TOP2ccs debulking that are dependent on the resolution of TOP2-DNA adducts by TDP2. This promotes the error free repair of TOP2-induced DSBs and enhances cells survival and virus production.

tetracycline-regulated BZLF1 were generated by retroviral transduction. Treatment with Dox resulted in comparable induction of the productive virus cycle assessed by the expression of early and late antigens and increased BPLF1 mRNA (S7A and S7B Fig). Analysis of DNA damage by γH2AX immunoblotting or immunofluorescence was inconclusive in this cell line pair due to the high levels of endogenous damage in TDP2$^{-/-}$ cells. Nevertheless, as observed in HEK293-EBV and LCL cells expressing wild type BPLF1, induction of the productive cycle was associated with significantly improved resistance to the toxic effect of Etoposide in the TK6 LCL, while the expression of BPLF1 had no effect in the subline lacking TDP2 (Fig 7A). Furthermore, EBV-DNA replication and virus production were significantly reduced in the KT6-TDP2$^{-/-}$ (Fig 7B). Collectively, these findings corroborate the conclusion that catalytically active BPLF1 supports cell survival and efficient virus production by promoting the non-proteolytic and TDP2-dependent debulking of TOP2ccs.

## Discussion

Although compelling evidence points to a pivotal role of topoisomerases in the replication of herpesviruses and other DNA viruses [9, 11, 12], very little is known about the mechanisms by

which the viruses harness the activity of these cellular enzymes. In this study, we have shown that the ubiquitin deconjugases encoded in the N-terminal domain of the EBV large tegument protein BPLF1 regulates the activity of TOP2 during productive EBV infection by promoting the proteasome-independent debulking of TOP2-DNA adducts. The findings highlight a previously unrecognized function of the viral enzyme in hijacking cellular functions that support cell survival and enable efficient virus production. Our proposed model for the regulation of TOP2 by BPLF1 is shown in Fig 7C.

We found that the viral DUB that is physiologically released from the large tegument protein during productive infection via caspase-1-mediated cleavage [32] selectively protects TOP2α and TOP2β from proteasomal degradation in cells treated with topoisomerase poisons, and is recruited to protein complexes containing TOP2α and TOP2β, effectively counteracting their ubiquitination (Figs 1 and 2A). The inhibition of TOP2 degradation was accompanied by stabilization of TOP2ccs (Fig 2B and 2C), which prevented the unmasking of TOP2-induced DSBs with consequent failure to activate the DDR (Fig 3). While important for cell survival, several lines of evidence suggest that the potent DDR triggered by the proteolytic debulking of TOP2ccs may also cause apoptosis and genomic instability in the surviving cells. This is because the protein-free DSBs unmasked by the degradation of TOP2 may engage multiple pathways for error-free or error-prone repair, including MRE11 nuclease-dependent homologous recombination (HR) [44] and non-homologous end joining (NHEJ) [46, 47]. Recent findings suggest that upon TOP2 degradation a substantial fraction of the Etoposide induced DSBs undergo extensive DNA end-resection [38], which favors mispairing and the occurrence of chromosomal rearrangements that compromise cell viability and promote genomic instability. The genotoxic effects were efficiently counteracted by inhibition of the proteasome prior or during Etoposide treatment, supporting the notion that a non-proteolytic resolution of TOP2ccs minimizes DSB misrepair and prevents genomic instability [38]. Here we have shown that expression of physiological levels of the catalytically active viral DUB during productive infection closely mimics the stabilization of TOP2ccs, inhibition of DDR activation and reduction of Etoposide toxicity observed upon inhibition of the proteasome.

Whilst in line with the notion that deubiquitination protects the substrate from proteasomal degradation, our findings point to the capacity of BPLF1 to shift the cellular strategy for TOP2cc debulking towards proteasome-independent pathways that ensures higher fidelity of DNA repair and preserve cell viability. In this context, it is important to notice that catalytically active BPLF1 did not affect the SUMOylation of TOP2cc in Etoposide treated cells (Fig 4A) and increased the relative abundance of SUMOylated TOP2β in productively infected LCLs (Fig 6E and 6F). Accumulating evidence suggest that SUMOylation may play multiple roles in the debulking of TOP2ccs. SUMOylation may promote the degradation of TOP2 by the proteasome, serving as a recognition signal for the SUMO-targeted ubiquitin ligases RNF4 [20], or other proteases, such as the SprT-family metalloproteases SPRTN [48] and ARC/GCNA [49] that exert a promiscuous role in the proteolytic debulking of DNA-protein adducts. Interestingly, the activity of SPRTN is inhibited by mono-ubiquitination [50] and recent findings suggest that depletion of the cellular SPRTN deubiquitinase USP11 leads to the accumulation of unrepaired DNA-protein adducts [51]. Although BPLF1 could potentially mimic the activity of the cellular DUB, our findings that BPLF1 stabilizes TOP2ccs in Etoposide treated and induced cells (Figs 1, 2, 5 and 6) excludes the involvement of SPRTN-mediated SUMO-dependent proteolysis.

Recent findings suggest that SUMOylation may also mediate the non-proteolytic resolution of TOP2ccs. The SUMOylation of DNA trapped TOP2 by the ZNF451 ligase was shown to promote a conformational change in the TOP2 dimer that allows the recruitment of TDP2 via a "split-SIM" SUMO2 engagement platform [23]. The conformational change grants direct

access of TDP2 to the tyrosyl-DNA covalent bond and promotes the error-free rejoining of the DSBs by the T4 DNA ligase [46] or by TOP2 itself [38]. While not directly addressing the role of SUMOylation in the resolution of TOP2ccs, our findings that TDP2 knockdown abolishes the capacity of physiological levels of BPLF1 to rescue the toxic effect of Etoposide in LCL cells (Fig 7A) and decreases the yield of infectious virus (Fig 7B) is consistent with a scenario where BPLF1 favors a proteasome-independent but TDP2-dependent resolution of covalent TOP2-DNA adducts, which allows the repair of TOP2-induced DSBs without activation of the DDR, promoting thereby cells survival and virus production.

Although the pivotal role of topoisomerases in both the latent and lytic replication of herpesviruses is firmly established [9, 52–54], the contribution of the individual enzymes is not well understood. Previous findings suggest that the torsion-relieving activity of TOP1 is essential for the *in vitro* reconstitution of HSV replication using purified viral proteins [55], and its recruitment to the viral replication complex at OriLyt is required for the replication of EBV [12] and KSHV [11]. Less is known about the function of the TOP2 isozymes, although their importance is underscored by the upregulation of TOP2 during the productive cycle of HCMV [53] and KSHV [11]. We have observed a strong downregulation of TOP2α mRNA and protein levels upon induction of the productive virus cycle in freshly established LCLs (Figs 6A and S6C) but not in EBV carrying HEK293 cells (Figs 5A and S6C). This discrepancy is likely to be explained by the findings that in LCLs the upregulation of BZLF1 is accompanied by a p53-dependent arrest of the cell cycle in G1/S [42], where p53 functions as a TOP2α transcriptional repressor [56]. Thus, by stabilizing p53, catalytically active BPLF1 may reinforce the BZLF1-p53 induced cell cycle arrest in LCLs with functional p53 but fail to do so in cells with mutated/inactivated p53, such as the adenovirus immortalized HEK293 or certain Burkitt's lymphomas. Indeed, we have previously shown that, in these cells, virus-production is associated with a BPLF1-deNEDDylase-dependent endoreduplication phenotype mediated by inactivation of nuclear Cullin ligases and stabilization of the licensing factor CDT1 [30].

In contrast to the cell-type-dependent regulation of TOP2α, the expression of TOP2β was either not affected or slightly upregulated in both productively infected HEK293 and LCLs, which is in line with the cell-cycle independent expression of this topoisomerase and its essential role in transcription [25]. Physiological levels of the active BPLF1 were sufficient to significantly reduce the levels of ubiquitinated TOP2β (Fig 6E) and promote the accumulation of TOP2βccs (Figs 5C and 6C), which correlated with inhibition of the DDR (Figs 5F and 6H) and resistance to the toxic effect of Etoposide (Figs 4C and 6I). Most importantly, TDP2 knockdown abolished the capacity of BPLF1 to rescue Etoposide toxicity (Fig 7A), supporting the conclusion that cell survival is dependent on the TDP2-mediated release of TOP2 from covalent DNA adducts and consequent DNA repair without activation of the DDR. TOP2-induced DSBs are likely to accumulate during EBV replication and transcription. Thus, the capacity of BPLF1 to selectively attenuate the TOP2-induced DDRs may contribute to extend the viability of cells entering the productive cycle and ensure efficient virus production. Of note, the salvage of TOP2β from proteolytic disruption is probably reinforced by the concomitant downregulation of the SUMO-dependent ubiquitin ligase RNF4 [43], which, together with the BPLF1-mediated deubiquitination, may ensure that sustained levels of TOP2 remain available throughout the productive virus cycle.

Although not essential for herpesvirus replication and infection, as also confirmed by our capacity to produce infectious virus and establish LCLs carrying the BPLF1^C61A mutant, the DUB activity of the large tegument proteins was shown to play important roles in the life cycle of these viruses by promoting efficient virus production [30, 57, 58] and by regulating various aspects of viral pathogenesis, including the establishment of latency [59], immune evasion [41, 60], and viral oncogenesis [61]. While documenting a previously unrecognized function of

BPLF1 during productive infection, our findings have interesting implications in the context of EBV-associated malignancies. Aberrant expression of BPLF1 mRNA in the absence of virus production has been reported in nasopharyngeal carcinoma, NK-T cell lymphomas, and a subset of gastric cancers [62, 63]. Conceivably, the capacity of BPLF1 to regulate the activity of TOP2 could provide a selective advantage to the rapidly proliferating malignant cells by promoting cell survival and limiting genomic instability. Etoposide and other topoisomerase poisons are used clinically as therapeutic anticancer agents against these malignancies [64]. Our data suggest that the expression of BPLF1 could serve as a biomarker to predict the effectiveness of chemotherapeutic regimens that incorporate topoisomerase poisons.

## Materials and methods

### Reagents

For a complete list of reagents, source and identifiers see S1 Table.

### Plasmids and recombinant lentivirus vectors

Eukaryotic expression vectors encoding the N-terminal domain of the EBV large tegument protein 3xFLAG-BPLF1 (amino acid 1–235) and the catalytic mutant BPLF1$^{C61A}$ [30] and the bacterial expression vector His-BPLF1 [65] were described previously. Lentiviral vectors encoding N-terminal 3xFLAG and V5 tandem tagged versions of BPLF1 aa 1–325 and the corresponding catalytic mutant BPLF1$^{C61A}$ under control of the doxycycline-inducible pTight promoter were produced by cloning the corresponding open reading frames into a modified version of the pCW57.1 plasmid (gift from David Root, Addgene plasmid #41393). The Gal1/10 His6 TEV Ura S. cerevisiae expression vector (12URA-B) was a gift from Scott Gradia (Addgene plasmid #48304) a plasmid expressing human TOP2α was kindly provided by the James Berger (John Hopkins School of Medicine, Baltimore, USA). The FLAG-TOP2α construct was created by in-frame cloning the 3xFLAG coding sequence (amino acids DYKDHDGDYKDHDIDYKDDDDKL) at the N-terminus of the TOP2α open reading frame. All cloning was performed using the ligation independent cloning protocol from the QB3 Macrolab at Berkeley (macrolab.qb3.berkeley.edu). A recombinant lentivirus vector expressing the coding sequence of the EBV transactivator BZLF1 under control of a tetracycline-regulated promoter was constructed by cloning the open reading frame amplified with the primers 5'-CGACCGGTATGATGGACCCAAACTCGAC-3' and 5'- CGACGCGTTTAGAAATTTAA GAGATCCTCGTGT-3' into the Age I and Mlu I sites of the pTRIPZ lentiviral vector (Thermo Fisher Scientific, USA). For virus production, HEK293FT cells were co-transfected with the pTRIPZ-BZLF1, psPAX and pMD2G plasmids (Addgene, Cambridge, MA) using JetPEI (Polyplus, Illkirch, France) according to the manufacture's protocol and cultured overnight in complete medium. After refreshing the medium, the cells were cultured for additional 48 h to allow virus release. Virus containing culture supernatant was briefly centrifuged and passed through a 0.45 μm filter to removed cell debris before aliquoting and storing at -80˚C for future use.

### Cell lines and transfection

HeLa cells (ATCC RR-B51S) and HEK293T (ATCC CRL3216) cell lines were cultured in Dulbecco's minimal essential medium (DMEM, Sigma-Aldrich), supplemented with 10% FBS (Gibco-Invitrogen) and 10 μg/ml ciprofloxacin (17850, Sigma-Aldrich) and grown in a 37˚C incubator with 5% $CO_2$. Stable HEK-rtTA-BPLF1/BPLF1$^{C61A}$ cell lines were produced by lentiviral transduction followed by selection in medium containing 2μg/ml puromycin for 2

weeks. Expression of FLAG-BPLF1/BPLF1$^{C61A}$ was induced by treatment with 1.5 μg/ml doxycycline and confirmed by anti-FLAG immunofluorescence and Western blot analysis. Clones expressing high levels of the transduced proteins were selected by limiting dilution. HeLa cells were transiently transfected with plasmids expressing FLAG-tagged version of BPLF1/BPLF1$^{C61A}$ using the lipofectamine 2000 (Invitrogen, California, USA) or jetPEI (Polyplus transfection, Illkirch FR) DNA transfection reagent according to the protocols recommended by the manufacturer. TK6 and TK6-TDP2$^{-/-}$ LCLs [44] were cultured in RPMI-1640 medium (R8758) supplemented with 10% horse serum (Gibco, 16050130) and 1 mM Sodium pyruvate (Thermo Fisher Scientific, 11360039). Sublines of the TK6 and TK6-TDP2$^{-/-}$ cells expressing a doxycycline-inducible BZLF1 transactivator were produced by culturing 10$^6$ cells with the recombinant lentivirus in presence of 8 μg/ml polybrene (TR-1003-G, Sigma-Aldrich) for 24 hours followed by replacement of the infection medium with fresh complete medium. The transduced cells were selected in medium containing 1 μg/ml (TK6) or 3 μg/ml (TK6-TDP2$^{-/-}$) puromycin for two weeks.

## Production of EBV immortalized lymphoblastoid cell lines (LCLs)

Peripheral blood mononuclear cells were purified from Buffy coats (Blood Bank, Karolinska University Hospital, Stockholm, Sweden) by Ficoll-Paque (Lymphoprep, Axis-shield PoC AS, Oslo, Norway) density gradient centrifugation, and B-cells were affinity-purified using CD19 microbeads (MACS MicroBeads, Miltenyi Biotec, Bergisch Gladbach, Germany) resulting in >95% pure B-cell populations. Infectious EBV encoding wild type or catalytic mutant BPLF1 were rescued from HEK293-EBV cells as previously described [41]. One million B-cells were incubated in 1 ml virus preparation for 1.5 h at 37˚C, followed by the addition of fresh complete medium and incubation at 37˚C in a 5% CO$_2$ incubator until immortalized LCLs were established. Sublines expressing a doxycycline-inducible BZLF1 transactivator were produced by culturing 10$^6$ LCL cells with the recombinant lentivirus in presence of 8 μg/ml polybrene (TR-1003-G, Sigma-Aldrich) for 24 hours followed by replacement of the infection medium with fresh complete medium. The transduced cells were selected in medium containing 0.8 μg/ml (LCL-BPLF1) or 0.25 μg/ml (LCL-BPLF1$^{C61A}$) puromycin for one or two weeks.

## Immunofluorescence

Transfected HeLa and HEK-rtTA-BPLF1/BPLF1$^{C61A}$ cells were grown on coverslips and induced with 1.5 μg/ml doxycycline for 24 h. For immunofluorescence analysis, the cells were fixed with 4% formaldehyde for 20 min, followed by permeabilization with 0.05% Triton X-100 in PBS for 5 min and blocking in PBS containing 4% bovine serum albumin for 40 min. After incubation for 1 h with primary antibodies and washing 3x5 min in PBS, the cells were incubated for 1 h with the appropriate Alexa Fluor-conjugated secondary antibodies, followed by washing and mounting in Vectashield-containing DAPI (Vector Laboratories, Inc. Burlingame, CA, USA). Images were acquired using a fluorescence microscope (Leica DM RA2, Leica Microsystems, Wetzlar, Germany) equipped with a CCD camera (C4742-95, Hamamatsu, Japan). Fluorescence intensity was quantified using the ImageJ software.

## Western blots

Cells were lysed in RIPA buffer (25 mM Tris-HCl pH 7.6, 150 mM NaCl, 1% Igepal, 1% sodium deoxycholate, 2% SDS) supplemented with protease inhibitor cocktail. Loading buffer (Invitrogen) was added to each sample followed by boiling for 10 min at 100˚C. The lysates were fractionated in acrylamide Bis-Tris 4–12% gradient gel (Life Technologies Corporation, Carlsbad, USA). After transfer to PVDF membranes (Millipore Corporation, Billerica, MA,

USA), the blots were blocked in TBS (VWR, Radnor, Pennsylvania, USA) containing 0.1% Tween-20 and 5% non-fat milk, and the membranes were incubated with the primary antibodies diluted in blocking buffer for 1 h at room temperature or over-night at 4°C followed by washing and incubation for 1 h with the appropriate horseradish peroxidase-conjugated secondary antibodies. The immunocomplexes were visualized by enhanced chemiluminescence (GE Healthcare AB, Uppsala, SE). For detecting topoisomerase-DNA adducts after treatment with topoisomerase poisons, the cells were lysed in alkaline buffer [36]. Briefly, cells treated for the indicated time with 5 μM Camptothecin or 80 μM Etoposide were lysed in 100 μl in buffer containing 200 mM NaOH, 2 mM EDTA, followed by the addition of 100 μl of 1M HEPES buffer (pH 7.4). Nucleic acids were removed by addition of 10 μl 100 mM CaCl2, 2 μl 1M DTT, and 200 U of micrococcal nuclease followed by incubation at 37°C for 20 min. Seventy μl of 4xLDS loading buffer (Invitrogen) were added to each sample followed by boiling for 10 min at 100°C before SDS-PAGE fractionation and western blot analysis.

## Immunoprecipitation and pull-down assays

Cells were harvested 48 h after transfection and lysed in NP40 lysis buffer (150 mM NaCl, 50 mM Tris-HCl pH7.6, 5 mM MgCl$_2$, 1 mM EDTA, 1% Igepal, 1 mM DTT, 10% glycerol) supplemented with protease/phosphatase inhibitor cocktail, 20 mM NEM and 20 mM Iodoacetamide for 30 min on ice. For immunoprecipitations under denaturing condition the lysis buffer was supplemented with 1% SDS followed by dilution to 0.1% SDS before IP. For BPLF1/ BPLF1$^{C61A}$ co-immunoprecipitation, the lysates were incubated for 3 h with 50 μl anti-FLAG packed agarose affinity gel (A-2220; Sigma) at 4°C with rotation. After washing 4 times with lysis buffer, the immunocomplexes were eluted with FLAG peptide (F4799; Sigma). For TOP2α and TOP2β immunoprecipitation, specific antibodies were added to cell lysates and incubated at 4°C for 3 h with rotation. The protein-antibody complexes were captured with protein-G coupled Sepharose beads (GE Healthcare) by incubation at 4°C for 1 h. The beads were washed 4 times with lysis buffer followed by boiling in 2xSDS-PAGE loading for 10 min at 100°C. The production of 6xHis-BPLF1 in bacteria and purification of the recombinant protein were done as previously described [65]. Recombinant human TOP2α was expressed and purified with the assistance of the Proteins Science Facility of the Karolinska Institutet according to a previously published protocol with slight modifications [66]. Briefly, URA-deficient yeast (kindly provided by Lena Ström, CMB Karolinska Institutet) were transformed with the TOP2α expression plasmid, grown initially in uracil-deficient media, then in YPLG (1% yeast extract, 2% peptone, 2% sodium DL-lactate, 1.5% glycerol) before induction of expression by addition of 2% galactose. The yeast was harvested by centrifugation and snap-frozen in liquid nitrogen. Proteins were extracted using a cryo-mill, and the filtered lysate was passed sequentially through HisTrap Excel nickel and HiTrap CP cation exchange columns (GE Healthcare) to purify the tagged TOP2α protein before incubation overnight with His-tagged TEV protease. The following day, the protein was passed through a HisTrap column to remove the cleaved His-tag and the TEV protease. The TOP2α protein was further purified on a Superdex 200 16/60 column, concentrated, and stored at -80°C. Relaxation and decatenation assays along with western blotting were performed to confirm protein purity and activity. Equimolar concentration of purified His-BPLF1 (0.35 μg) and FLAG-TOP2α (2 μg) were incubated in binding buffer (100 mM NaCl, 50 mM Tris-HCl, 1 mM DTT, 0.5% Igepal) for 20 min at 4°C. Anti-FLAG agarose affinity gel (A-2220; Sigma) or Ni-NTA beads (Qiagen) were added followed by incubation for 60 min or 20 min at 4°C with rotation. The beads were extensively washed, and bound proteins were eluted with FLAG peptide or 300 mM imidazole in buffer containing 50 mM Tris-HCl pH 7.6, 50 mM NaCl and 1 mM DTT.

### Rapid approach to DNA adducts recovery (RADAR) assay

TOP2ccs were isolated by RADAR assays as described [34]. Briefly, cells cultured in 6 well plates were treated with 80 µM etoposide for 30 min or 4 h and then lysed in 800 µl DNAzol. Following the addition of 400 µl absolute ethanol, the lysates were cooled at -20˚C and then centrifuged at 14000 rpm for 20 min at 4˚C. After repeated washing in 75% ethanol the nucleic acid pellets were dissolved in 100 µl $H_2O$ at 37˚C for 15 minutes, followed by treatment with 100 µg/ml RNaseA. The concentration of DNA was measured and 10 µg DNA from each sample were treated with 250 U micrococcal nuclease supplemented with 5 mM $CaCl_2$ before the addition of loading buffer and detection of trapped protein by western blot.

### Reverse transcription and real-time PCR

Total RNA was isolated using the Quick-RNA MiniPrep kit (Zymo Research, Irvine, CA, USA) with in-column DNase treatment according to the instructions of the manufacturer. One microgram of total RNA was reverse transcribed using SuperScript VILO cDNA Synthesis kit (Invitrogen). PCR amplification was performed with the LC FastStart DNA master SYBR green I kit in a LightCycler 1.2 instrument (Roche Diagnostic) using the primers listed in S2 Table. The PCR reactions were as follows: denaturation at 95˚C for 10 min, followed by 40 cycles at 95˚C for 8 sec, 60˚C for 5 sec, 72˚C for 8 sec. The relative levels of mRNA were determined from the standard curve using the endogenous MLN51(Metastatic Lymph Node 51) gene as reference.

### EBV DNA replication and release of infectious virus

Virus replication and the release of infectious virus were monitored in cell pellets and culture supernatants three days after induction with 1.5 µg/ml Doxycycline. Culture supernatants were cleared of cell debris by centrifugation of 5 min at 14000 rpm and treated with 20 U/ml DNase I (Promega, Madison, WI, USA) to remove free viral DNA. Viral DNA was isolated using the DNeasy Blood & Tissue Kit (Qiagen, Hilden, Germany) and quantitative PCR was performed as described above with primers specific for a unique sequence in EBNA1: 5'-GG CAGTGGACCTCAAAGAAG-3', 5'-CTATGTCTTGGCCCTGATCC-3' and the cellular EF1α (Elongation factor 1α) 5'-CTGAACCATCCAGGCCAAAT-3', 5'GCCGTGTGGCA ATC CAAT-3' as reference. Virus replication was calculated as the ratio between the amount of viral DNA in induced versus untreated cells.

### MTT assay

For assay of cell viability, $2x10^4$ HEK-rtTA-BPLF1/BPLF1$^{C61A}$, $5x10^4$ LCL-EBV-BPLF1/BPLF1$^{C61A}$ or $5x10^4$ TK6/TK6-TDP2$^{-/-}$ were plated in 150 µl medium in triplicate wells of a 96 well plate without or with the addition of the indicated concentrations of Etoposide. After incubation for 20 h at 37˚C in a 5% $CO_2$ incubator, 50 µl culture medium containing 1 mg/ml Methylthiazolyldiphenyl-tetrazolium bromide (MTT, M5655, Sigma-Aldrich) were added to the wells followed by incubation for additional 4 h. The MTT formazan crystals produced by mitochondrial dehydrogenases in living cells were solubilized by the addition of 50 µl 10% SDS and O.D. was measured at 540 nm in a plate reader. Relative viability was calculated after subtraction of the background O.D. of media alone.

## Supporting information

**S1 Table. Reagents used in the paper.**
(DOCX)

**S2 Table. qPCR primers used in the paper.**
(DOCX)

**S1 Fig. Characterization of the HEK-rtTA-BPLF1/BPLF1-C61A cell lines.** A) Expression of BPLF1 was detected in western blots of cells treated for 24 h with the indicated amount of Dox using antibodies to the FLAG and V5 tags. Induction of 24 h with 1.5 μg/ml Dox was used in all subsequent experiments. B) Representative micrographs illustrating the expression of BPLF1/BPLF1C61A in untreated and Dox-treated cells. Confocal images were obtained at 40x lens objective magnification. BPLF1 is in green and cell nuclei were stained with DAPI (blue). Strong FLAG fluorescence was regularly detected in approximately 50% of the induced cells.
(TIF)

**S2 Fig. BPLF1 does not interact with TOP1 and recombinant BPLF1 binds to TOP2.** (A) HEK293T cells were transfected with FLAG-BPLF1, FLAG-BPLF1$^{C61A}$, or empty FLAG-vector and then treated with 40 μM Etoposide for 30 min. Cell lysates were immunoprecipitated with anti-FLAG conjugated agarose beads and western blots were probed with the indicated antibodies. TOP2α was readily detected in the immunoprecipitates while TOP1 was consistently absent. Representative western blots from one of two independents experiments giving similar results are shown. (B,C,D) The interaction of yeast expressed FLAG-TOP2α or TOP2α lacking the C-terminal domain (FLAG-TOP2α-ΔCTD) with bacterially expressed His-BPLF1 was assayed in pull-down assays. Equimolar amounts of the proteins were mixed and FLAG (B, D) or Ni-NTA (C) pull-downs were probed with antibodies specific for FLAG or His tags. A weak interaction of BPLF1 with TOP2α was detected independently of the presence of the TOP2α C-terminal domain. Western blots from one representative experiment out of two are shown in the figure.
(TIF)

**S3 Fig. BPLF1 does not affect the induction of TOP2ccs in cells treated by Etoposide.** HEK-rtTA-BPLF1 cells were cultured for 24h in the presence or absence of Dox and then treated for 15 min with the indicated concentrations of Etoposide. TOP2ccs were isolated according to the RADAR protocol and blotted on nitrocellulose paper using a dot blot apparatus followed by probing with antibodies specific for TOP2α and TOP2β. Comparable amounts of TOP2ccs were detected at each Etoposide independently on BPLF1 expression. One representative experiment out of two is shown.
(TIF)

**S4 Fig. BPLF1 inhibits the resolution of TOP2ccs.** (A) HEK-rtTA-BPLF1 cells were cultured with or without 1.5 mg/ml Dox for 24 h and then treated with 80 μM Etoposide alone or together with 10 μM MG132. Cells harvested after 1 h or 6 h were lysed in alkaline buffer and the formation of TOP2cc was investigated by probing western blots with the TOP2β antibody. The TOP2cc are visualized as smears of DNA cross-linked TOP2β above the main band. Probing with the anti-TOP1 antibody confirmed the selective induction of TOP2bcc in Etoposide treated cells. GAPDH was used as the loading control. Western blots from one representative of three independent experiments are shown in the figure. (B) Densitometry quantification confirming the stabilization and TOP2βcc in BPLF1 expressing cells. The mean ± SE of two independent experiments is shown. **P<0.01.
(TIF)

**S5 Fig. Transfection of catalytically active BPLF1 inhibits activation of the DDR and DNA repair in etoposide treated HeLa cells.** HeLa cells transiently transfected with plasmids

expressing FLAG-BPLF1/BPLF1-C61A were treated for 6 h with 40 μM etoposide before fixation and staining with the indicated antibodies. Representative micrographs of cells co-stained with antibodies to FLAG, the DNA-DSB marker γH2AX, and the DNA repair markers 53BP1 and BRCA1. Expression of catalytically active BPLF1 was associated with decrease γH2AX and BRCA1 fluorescence and failure to accumulate 53BP1 foci. Images from one representative experiment out of three are shown. The intensity of γH2AX and BRCA1 fluorescence and the number of cells showing ≥2 53BP1 foci were quantified in BPLF1 positive and negative cells from the same transfection experiment using the ImageJ software. Mean ± SE of two or three independent experiments where a minimum of 50 BPLF1 positive and 50 BPLF1 negative cells was scored in each condition.
(TIF)

**S6 Fig. Induction of the productive virus cycle and quantification of TOP2 mRNA in cells carrying recombinant EBV expressing wild type and mutant BPLF1.** The productive cycle was induced in HEK293-EBV-BPLF1/BPLF1$^{C61A}$ and LCL-EBV-BPLF1/BPLF1$^{C61A}$ by culture for 72 h in the presence of 1.5 μg/ml Dox. (A) Representative western blots of total cells lysates from Dox treated and untreated cells probed with antibodies to the immediate early antigen BZLF1, the early antigen BMRF1 and the late antigen BFRF3. The expression of BPLF1 (B), TOP2α and TOP2β (C) mRNA was quantified by qPCR. The mean ± SE fold increase relative to uninduced controls recorded in three independent experiments is shown in the figures.
(TIF)

**S7 Fig. Induction of the productive virus cycle in the TK6/TK6-TDP2$^{-/-}$ cell lines.** The productive cycle was induced in the TK6 and TK6-TDP2$^{-/-}$ LCLs expressing tetracycline regulated BZLF1 by culture for 72 h in the presence of 1.5 μg/ml Dox. (A) Representative western blots of total cells lysates from Dox treated and untreated cells probed with antibodies the immediate early antigen BZLF1, the early antigen BMRF1 and the late antigen BFRF3. GAPDH was used as loading control. One representative experiment our of 4 is shown. (B) BPLF1 mRNA was quantified by qPCR. The mean ± SE fold increase relative to uninduced controls recorded in three independent experiments is shown.
(TIF)

## Acknowledgments

We are deeply grateful to Dr Arne Lindqvist and Prof Nico Dantuma (CMB, Karolinska Institutet) for generously sharing knowledge and reagents and to Ms. Lisa Wohlgemuth (Ulm University, Germany) for technical assistance.

## Author Contributions

**Conceptualization:** Laura Baranello, Maria G. Masucci.

**Formal analysis:** Jinlin Li, Noemi Nagy, Maria G. Masucci.

**Funding acquisition:** Maria G. Masucci.

**Investigation:** Jinlin Li, Noemi Nagy, Jiangnan Liu, Soham Gupta, Teresa Frisan, Thomas Hennig, Donald P. Cameron.

**Methodology:** Noemi Nagy, Soham Gupta, Laura Baranello.

**Project administration:** Maria G. Masucci.

**Supervision:** Maria G. Masucci.

**Validation:** Jinlin Li.

**Visualization:** Jinlin Li, Maria G. Masucci.

**Writing – original draft:** Jinlin Li, Noemi Nagy, Jiangnan Liu, Soham Gupta, Teresa Frisan, Donald P. Cameron.

**Writing – review & editing:** Laura Baranello, Maria G. Masucci.

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
