## [Decision Letter · Decision Letter 0]

28 Jul 2021

Dear Prof. Masucci,

Thank you very much for submitting your manuscript "The Epstein-Barr virus deubiquitinating enzyme BPLF1 regulates the activity of Topoisomerase II during productive infection" for consideration at PLOS Pathogens. As with all papers reviewed by the journal, your manuscript was reviewed by members of the editorial board and by several independent reviewers. In light of the reviews (below this email), we would like to invite the resubmission of a significantly-revised version that takes into account the reviewers' comments.

Three reviewers found the work interesting, but request major revisions. Reviewer 1 has requested a repeat of Fig 5E or 3 using TK6 cell pair with or without TP2. Reviewer 2 requests that BPLF1 be assayed in cells supporting the full lytic cycle and a test for direct interaction between BPLF1 and TOP2. Reviewer 3 requests some clarifications and a request to mutate TOP2 sumoylation sites, if plausible..

We cannot make any decision about publication until we have seen the revised manuscript and your response to the reviewers' comments. Your revised manuscript is also likely to be sent to reviewers for further evaluation.

Sincerely,

Paul M Lieberman

Associate Editor

PLOS Pathogens

Blossom Damania

Section Editor

PLOS Pathogens

Kasturi Haldar

Editor-in-Chief

PLOS Pathogens

orcid.org/0000-0001-5065-158X

Michael Malim

Editor-in-Chief

PLOS Pathogens

orcid.org/0000-0002-7699-2064

Three reviewers found the work interesting, but request major revisions. Reviewer 1 has requested a repeat of Fig 5E or 3 using TK6 cell pair with or without TP2. Reviewer 2 requests that BPLF1 be assayed in cells supporting the full lytic cycle and a test for direct interaction between BPLF1 and TOP2. Reviewer 3 requests some clarifications and a request to mutate TOP2 sumoylation sites, if plausible..

Reviewer's Responses to Questions

**Part I - Summary**

Reviewer #1: In this interesting manuscript, Li and colleagues describe a role of the EBV deubiquitinase (DUB) function embedded in the tegument protein BPFL1 in the regulation of the cellular topoisomerases TOP2alpha and TOP2beta during the lytic replication cycle. This is an important observation that contributes significantly to our understanding of the role of this viral deubiquitinase during the productive replication cycle of EBV and adds to our appreciation of the broad range of cellular proteins targeted by this viral enzyme. The experiments are of high quality and the results support the conclusions made by the authors. The authors should discuss the importance of their findings in the context of the observation that the DUB function of BPFL1 is not essential for productive EBV replication, as reported by others. It would also be very informative if they could add an experiment that directly links the Tyrosyl-DNA-phosphodiesterase to the regulation of DNA damage by BPFL1.

Reviewer #2: The manuscript by Li et al. provides an interesting mosaic of robust biochemical and cellular data on a presumed and previously unknown function of the N-terminal part of the large EBV tegument protein BPLF1. The manuscript focuses on induced or transiently transfected and thus ectopically expressed proteins and their interactions. Together, the single pieces of evidence form a compelling image of an interesting biochemical pathway in which BPLF1 prevents cellular DDR which, in turn, can result in an antiviral response. While most of the presented data are fine and are nicely done, the manuscript falls short in studying robust models of viral production. Instead, the manuscript relies on an etoposide induced DDR, which has nothing to do with a cell that supports de novo synthesis of a herpesvirus such as EBV. To have a meaningful story it appears essential that some of the key experiments are repeated such that virus progeny can be quantified as a function of modulating levels of BPLF1, TOP2alpha and beta, and TDP2. The number of released virions matter in a viral model and is THE relevant parameter and benchmark.

Reviewer #3: In this manuscript, Li et al investigated the regulation of Topoisomerase-II (TOP2) by an EBV-encoded protease called BPLF1. BPLF1 is a big tegument protein that catalyzes the deconjugation of ubiquitin- and NEDD8-modified proteins. The authors discovered that BPLF1 can interact TOP2 and trigger its de-ubiquitination in cells treated with TOP2 inhibitor and during viral replication. Consequently, this led to the stabilization of TOP2 trapped in cleavage complexes. They demonstrated that this phenotype is lost in cells expressing catalytic dead BPLF1. They further highlighted the role of tyrosyl-DNA phosphodiesterase 2 (TDP2) in control of BPLF1-mediated release of DNA-trapped TOP2, DNA repair and cell survival. The manuscript contains many high-quality western blot images.

Overall, this is an interesting manuscript that links BPLF1, de-ubiquitination and TOP2 to EBV replication and cell survival.

**Part II – Major Issues: Key Experiments Required for Acceptance**

Reviewer #1: 1. In figure 6 the authors show that the cellular Tyrosyl-DNA-phosphodiesterase TP2 is required for the ability of BPFL1 to moderate etoposide cell death. Based on results shown in other figures, in particular figure 5, they assume that this role of TP2 in preventing etoposide-induced cell death is due to an increased DNA damage seen in the absence of TP2, when the lytic cycle is induced (Fig. 5E). It would be very informative, if they could show that TP2 really does impact on DNA damage, by repeating the experiment shown in Fig. 5E or in Fig. 3 with the TK6 and TK6-TP2 -/- pair of LCL cell lines.

Reviewer #2: It is mandatory to investigate the consequences of BPLF1 expression, deubiquitination of TOP2 alpha and beta and the role on TDP2 in cells that support EBV’s full lytic phase. It is necessary to study the outcome of manipulating the expression levels of all components in question (BPLF1, TOP2alpha and beta – depending on the cellular model – and TDP2) in cell lines such as Akata cells and/or HEK293 cells engineered to carry EBV and EBV mutant derivatives capable of releasing infectious virus in principle. Measuring some levels of selected viral proteins in lytically induced cells (that do not support the full lytic phase of EBV) is not convincing. BPLF1, as a tegument protein is a component of EB virions and thus could play a role during the entire lytic phase. Modulation of the expression of the relevant players can be achieved by shRNA technologies or even CRISPR-Cas9 or any other experimental approach.

Whether BPLF1 and TOP2 physically interact remains uncertain, although this is a very critical question. It appears necessary to do co-immunoprecipitation experiments in cells, which have been induced to undergo the lytic phase of EBV using endogenous protein levels.This is critical because some of the data suggest that this interaction is indirect (Fig. S2B,C). It is difficult to understand how topoisomerases, which make direct contact with DNA and form trapped in-cleavage-complexes can be deubiquitinated by BPLF1 if the viral enzyme does not even make direct physical contact with its ubiquitinated substrate. In fact, it should be possible to do global ChIP experiments to demonstrate the binding of BPLF1 to (viral!) DNA via TOP2.

Reviewer #3: Specific questions,

1. Line 254 “expose the tyrosyl-DNA bond to the activity of Tyrosyl-DNA-phosphodiesterase-2 (TDP2)” is this referring to “expose the tyrosyl-DNA bond to the active sites of Tyrosyl-DNA-phosphodiesterase-2 (TDP2)”

2. Line 264-266 and Figure 4: The downregulation of TOP2 ubiquitination does not affect its SUMOylation level. Does this mean SUMOylation and ubiquitination occur on different sites? What happens if the TOP2 SUMOylation sites are mutated? Could de-ubiquitination of BPLF1 still has similar phenotype? The molecular mechanism for this part is less clear.

3. Figure 4C. “HEK-EBV-BPLF1/BPLF1-C61A -Dox”: is this referring to a mixture of two different cell lines? If yes, why not separate them as the ones treated with Dox. Similarly question for Figure 6G. “LCL-BPLF1/BPLF1-C61A -Dox” and Figure 6H “TK6/TK6-TDP2-/-”.

**Part III – Minor Issues: Editorial and Data Presentation Modifications**

Reviewer #1: 1. Fig. 2a: How reliable is the observation that, in the presence of BPLF1, there is less ubiquitination of TOP2a and TOP2b, if there is also an overall reduction in ubiquitinated proteins (input)? Later on in the manuscript (Fig. 6) the authors show that, under physiological levels of BPFL1 expression in EBV-infected cells, this overall reduction of ubiquitination of cellular proteins does not occur, while BPFL1 still reduces the ubiquitination of TOP2b. Therefore, at least in EBV-infected B cells, this point regarding Fig. 2a is not a crucial one, but the authors might want to point this out when discussing the results shown in Fig. 2a.

2. Fig. S3: In the absence of MG132, the difference in TOP2b levels between Dox-treated and untreated cells is most pronounced at 1 hour, rather than 6 hours of etoposide treatment, suggesting that the impact of BPFL1 is stronger at the stage of formation of DNA-TOP2 adducts than during their resolution. Similarly, in Fig. 2B the TOP2a band in etoposide-only treated BPFL1-expressing cells is already stronger than in BPFL1-negative cells at 0.5 hours, again suggesting an effect of BPFL1 on the formation rather than the resolution of TOP2a-DNA adducts. Also, in Fig. 4A, the effect of BPFL1 on the etoposide-induced ubiquitination of TOP2ccs is already seen after 30 min. Can the authors therefore be confident, that BPFL1 affects the resolution rather than the formation of TOP2ccs or might it be better to discuss this point more cautiously?

3. In spite of the undoubted ability of the BPFL1 DUB function to modulate the ubiquitination of a range of cellular proteins, this DUB activity, which is conserved among herpesviruses, has not been sown to be essential for viral productive replication. The authors should therefore address this conundrum in their discussion to put the importance of their findings in the appropriate context. Maybe their observation that in LCLs TOP2a is downregulated at the transcriptional level during lytic reactivation may provide a partial explanation. It would then be interesting to discuss/speculate in which context (LCL proliferation? Lytic replication of EBV in p53-competent epithelial cells?) the role of BPFL1 in the regulation of TOP2a and TOP2b might be particularly important as a basis for designing experiments that would test an EBV BAC mutant with an inactivated DUB in the most relevant context.

Reviewer #2: - Irregular usage of comma instead of dot in numbers

- It is highly irritating to see that ectopic expression of wildtype BPLF1 leads to a global loss of ubiquitinated proteins in cells as in Fig. 2A, bottom panel. Needs to be addressed and controls should be included that demonstrate the specificity of deubinquitination of TOP2.

- Fig. 3, middle panels, BPLF1 (wildtype), dox+, Eto: panel B indicates a dramatic loss of 53BP1 foci in doxycycline-induced cells but the microscopic image in panel A does not show this. Needs to corrected or the visualization should be improved.

- Fig. 6 panel H: TK6/TK6-TP2-/- - Dox should read TK6/TK6-TDP2-/- - Dox

Reviewer #3: Minor,

1. Figure 1C, 2C, 4B-C, 6B, 6G-: the number labeling should be corrected, eg. “0,01” should be “0.01”

2. Figure 3B, the labeling of “– and +” is confusing. It should show as BPLF1-, BPLF1+ etc.

3. Figue 5E, BPLF-1C61A should be BPLF1-C61A

4. Line 871, “HEK-rtTA-BPLF1/BPLFC61A ” should be “HEK-rtTA-BPLF1/BPLF1C61A ”

5. Table S1 “(van Gent, M., et al. 2024)” this reference is not included in the manuscript. Also the year 2024 might be a typo.

6. Table S1. “HEL293-EBV-BPLF1/BPLF1-C61A”: first, typo on HEL; second, is this referring to one cell line or two different cell lines? It’s better to make this clear in the table.

7. Table S1. “TK6/TK6-TDP2-/-”: similarly, is this referring to one cell line or two different cell lines?

PLOS authors have the option to publish the peer review history of their article (what does this mean?). If published, this will include your full peer review and any attached files.

Reviewer #1: No

Reviewer #2: No

Reviewer #3: No
---

## [Decision Letter · Decision Letter 1]

11 Sep 2021

Dear Prof. Masucci,

We are pleased to inform you that your manuscript 'The Epstein-Barr virus deubiquitinating enzyme BPLF1 regulates the activity of Topoisomerase II during productive infection' has been provisionally accepted for publication in PLOS Pathogens.

Best regards,

Paul M Lieberman

Associate Editor

PLOS Pathogens

Blossom Damania

Section Editor

PLOS Pathogens

Kasturi Haldar

Editor-in-Chief

PLOS Pathogens

orcid.org/0000-0001-5065-158X

Michael Malim

Editor-in-Chief

PLOS Pathogens

orcid.org/0000-0002-7699-2064

Reviewer Comments (if any, and for reference):

Reviewer's Responses to Questions

**Part I - Summary**

Reviewer #1: In this very interesting manuscript, Li et al. describe a role for the DUB moiety in the EBV tegument protein BPLF1 for the regulation of topoisomerase 2b during the lytic replication cycle. The authors convincingly show that BPLF1 can deubiquitinate and thereby stabilise Top 2b, thereby subduing DNA damage in EBV-replicating cells and aiding the survival of EBV-infected cells and with it, virus production. This is an important observation that links BPFL1 to the DNA damage pathway. It also extends the list of viral and cellular proteins that are deubiquitinated and thereby regulated by BPLF1, thereby adding to an emerging picture of a pleiotropic role of this viral enzyme.

The authors have addressed all the reviewers' comments in a satisfactory manner by additional experiments, and clarifying and/or discussing unclear issues.

Reviewer #2: The authors have adequately responded to my different points of critique (both major and minor), they added additional experimental data and figure panels in the revision process such that my concerns have been satisfactorily addressed.

Reviewer #3: The authors addressed my questions and made a valid rebuttal.

**Part II – Major Issues: Key Experiments Required for Acceptance**

Reviewer #1: no further experiments required

Reviewer #2: none

Reviewer #3: (No Response)

**Part III – Minor Issues: Editorial and Data Presentation Modifications**

Reviewer #1: 1. Line 179: three words are crossed out and should be removed from the final version of the manuscript.

2. Do the authors want to clarify in the discussion, whether they interpret the results shown in figure 7b (reduced virus production in TDP2 -/- cells compared to cells with a functional TDP2) as being the consequence of the increased rates of cell death seen in the former compared to the latter after induction of BPLF1 (Fig. 7a)? Or do they interpret the result shown in figure 7b as the result of increased DNA damage in the viral genome in the absence of TDP2?

Reviewer #2: none

Reviewer #3: (No Response)

PLOS authors have the option to publish the peer review history of their article (what does this mean?). If published, this will include your full peer review and any attached files.

Reviewer #1: **Yes: **Thomas F. Schulz, MD

Reviewer #2: No

Reviewer #3: No

---

## [Editor Report · Acceptance letter]

16 Sep 2021

Dear Prof. Masucci,

We are delighted to inform you that your manuscript, "The Epstein-Barr virus deubiquitinating enzyme BPLF1 regulates the activity of Topoisomerase II during productive infection," has been formally accepted for publication in PLOS Pathogens.

Best regards,

Kasturi Haldar

Editor-in-Chief

PLOS Pathogens

orcid.org/0000-0001-5065-158X

Michael Malim

Editor-in-Chief

PLOS Pathogens

orcid.org/0000-0002-7699-2064